# Selective Rotary Position Embedding

**Sajad Movahedi**[*,1,4], **Timur Carstensen**[*,1,3], **Arshia Afzal**[*,2],
**Frank Hutter**[1,3,5], **Antonio Orvieto**[†,1,4], **Volkan Cevher**[†2]
Equal contribution[*], Equal supervision[†]
ELLIS Institute Tübingen[1], LIONS, EPFL[2], University of Freiburg[3],
Max-Planck-Institute for Intelligent Systems[4], Prior Labs[5]
`sajad.movahedi@tue.ellis.eu`   `timurcarstensen@gmail.com`
`arshia.afzal@epfl.ch`

## Abstract

Positional information is essential for language modeling. Softmax Transformers with Rotary Position Embeddings (RoPE) encode it with fixed-angle rotations, while linear Transformers rely on input-dependent gates that only decay past key-value norms. We provide a theoretical argument for the necessity of a rotation and decay component in well-performing sequence models, and observe that the missing ingredient in linear models is precisely the rotation that softmax attention performs implicitly. We introduce Selective Rotary Position Embedding (*Selective RoPE*), an input-dependent, learnable rotary embedding that generalizes RoPE to arbitrary angles and composes seamlessly with decay gates. Equipping gated linear attention with *Selective RoPE* yields a complex-valued recurrent layer that can be implemented efficiently with the RoPE trick. On synthetic benchmarks (MQAR, copying, state tracking) and 370M-parameter language-model pre-training, the method improves recall, downstream accuracy, and expressivity while adding minimal architectural overhead. We open-source our implementation here.

## 1 Introduction

Transformers with softmax attention (Vaswani et al., 2017) form the foundation of state-of-the-art language models, largely because every token can attend to all past tokens without decay, yielding strong in-context recall. This expressivity comes at a computational cost: even with memory-efficient kernels, the arithmetic complexity of softmax attention is quadratic in the sequence length. To mitigate this, a parallel line of work has developed sub-quadratic sequence models. These models are recurrent architectures that run in *linear* time and require only *constant* memory per step at inference (Katharopoulos et al., 2020; Yang et al., 2024b; Gu & Dao, 2023; Dao & Gu, 2024). Their main bottleneck is the fixed state size: information must be selectively retained or overwritten, which often harms long-horizon retrieval. Consequently, recent progress has focused on improving state management in these models.

Theoretical investigations have proven the benefits of complex gating mechanisms for this purpose (Orvieto et al., 2023b; Ran-Milo et al., 2024; François et al., 2025). However, in practice selective gating (Yang et al., 2024a; Gu & Dao, 2023; Dao & Gu, 2024), more expressive state updates (Yang et al., 2024b; Siems et al., 2025; Peng et al., 2025) and readouts (Peng et al., 2025; Hu et al., 2025) usually play the main role in memory management. As these mechanisms largely act by modulating the *norms* of key-value associations (i.e., how quickly they decay) and do not directly provide the complementary capability of *rotating* query-key representations to encode relative position, they fail to materialize the full benefits of the gating mechanism as done in earlier variants of recurrent sequence models (Gu et al., 2022c; Orvieto et al., 2023a; Sun et al., 2023b).

**Our view: recall needs rotation *and* decay.** We posit that strong recall requires two complementary mechanisms: (i) *rotation*, to encode relative position while preserving norms, and (ii) *decay*, to selectively discard past keyvalue associations. By relying on the relationship between the softmax function and Random Fourier Features (RFF), we reveal that softmax attention can be interpreted as performing *input-dependent selective rotations* of query-key pairs, thus motivating the use of RoPE-like rotation matrices in recurrent models.

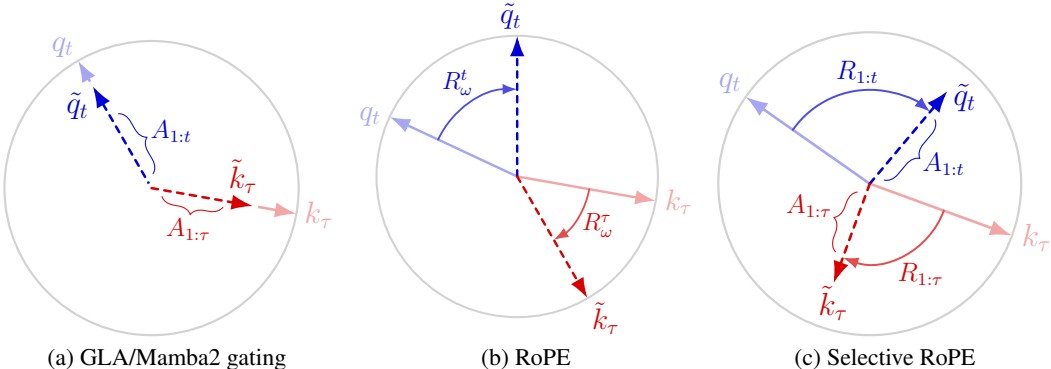

Figure 1: **Scaling vs. rotation on the unit circle.** *Left:* GLA/Mamba-style forgetting encodes history primarily through *norm scaling*: cumulative gates $A_{1:t}$ and $A_{1:\tau}$ attenuate $q_t$ and $k_\tau$ to $\tilde{q}_t$ and $\tilde{k}_\tau$ without changing direction. *Middle:* RoPE encodes position via *fixed* index-dependent rotations ($R_\omega^t$, $R_\omega^\tau$) that preserve norms. *Right:* Selective RoPE + decay composes *input-dependent rotations* with *norm scaling*, capturing both phase (relative position) and forgetting.

**Why rotation alone is insufficient.** A purely complex (rotation-only) linear recurrent model behaves like a spectral analyzer with fixed state size. When applied to the input sequence, which can be seen as a finite-length signal, such a model suffers from spectral leakage, leading to a poorer approximation of the underlying signal. This can be mitigated by adding an exponentially decaying component to the recurrence. In linear recurrent architectures, the analog of spectral leakage is the suboptimal compression of keyvalue associations into the fixed-size hidden state, which is remedied by adding *selective gating* to the state transition.

Based on our recipe, we introduce a complex version of Gated Linear Attention (GLA) (Yang et al., 2024a) and demonstrate its superior performance and expressivity. In practice, we show that by using the RoPE trick (Su et al., 2021), we are able to efficiently compute a complex GLA by applying a learned, input-dependent rotary position embedding to the queries and keys. ***Selective RoPE is easily incorporated into most gated linear Transformers by applying input-dependent and learned rotations to queries and keys.*** In Figure 1, we provide a visual comparison.

**Contributions.**

- **Unifying view.** We show that effective recall needs both *rotation* and *decay*. Softmax implicitly implements input-dependent rotations. Complex-only linear models suffer from spectral leakage, motivating explicit decay. Real parts forget; imaginary parts encode position.

- **Theory.** (1) Using an RFF approximation of the exponential kernel, we expose selective rotations in softmax and derive an optimal temperature distribution that matches exponentially decaying frequencies used in RoPE. (2) We investigate diagonal SSMs as spectral analysers, showing that they suffer from spectral leakage which can be suppressed through forget gates.

- **Method:** We propose ***Selective RoPE***. An input-dependent rotary embedding that generalizes RoPE to learned angles and composes with gates; we make the implementation efficient by incorporating the RoPE trick in both linear and softmax attention.

- **Empirics.** We integrate *Selective RoPE* with GLA, significantly boosting performance on recall-centric synthetic tasks (MQAR, copying, state tracking) and improving downstream language modeling.

## 2 BACKGROUND

In this section, we provide some mathematical background, beginning with an introduction of the Transformer architecture, its relevant variants and the RoPE trick (Su et al., 2021). The results of this section also establish a relationship between a linear Transformer architecture with RoPE and a recurrent sequence model with rotations in the state transition. This relationship later aid us through a more efficient computation of the latter.

**Transformers.** Standard causal softmax attention (Vaswani et al., 2017) transforms a sequence of $L$ inputs $(x_t)_{t=1}^L$ into the sequence of outputs $(o_t)_{t=1}^L$, with $x_t, s_t, o_t \in \mathbb{R}^d$ and $z_t \in \mathbb{R}$:

$$o_t = \frac{s_t}{z_t}, \quad s_t = \sum_{\tau=1}^t \exp\left(\tfrac{1}{\sqrt{d}} q_t^\top k_\tau\right) \cdot v_\tau, \quad z_t = \sum_{\tau=1}^t \exp\left(\tfrac{1}{\sqrt{d}} q_t^\top k_\tau\right), \quad (1)$$

where $q_t, k_t, v_t = W_q x_t, W_k x_t, W_v x_t$, and $W_q, W_k, W_v \in \mathbb{R}^{d \times d}$ are the projection matrices and $z_t$ is the normalization factor. Linear attention (Katharopoulos et al., 2020) replaces the exponential kernel in softmax attention with a kernel with a positive feature map $\phi(\cdot) : \mathbb{R}^d \to (\mathbb{R}^+)^d$, which gives rise to the following model:

$$o_t = \frac{S_t \phi(q_t)}{z_t^\top \phi(q_t)}, \quad S_t = \sum_{\tau=1}^t v_\tau \phi(k_\tau)^\top, \quad z_t = \sum_{\tau=1}^t \phi(k_\tau). \quad (2)$$

Here $S_t \in \mathbb{R}^{d \times d}$ and $z_t \in \mathbb{R}^d$ are the state and the normalization factor. Due to the linear relationship, one can write the hidden state and the normalization factor in a recurrent form as: $S_t = S_{t-1} + v_t \phi(k_t)^\top$ and $z_t = z_{t-1} + \phi(k_t)$. Moving forward, we subsume the feature map $\phi(\cdot)$ into query-key vectors to simplify notation and drop the normalization factor $z_t$ following Sun et al. (2023a). Equation (2) was enhanced with a *forget gate*, $A_t$, to manage the finite-sized hidden state better when processing long sequences:

$$S_t = S_{t-1} A_t + v_t k_t^\top, \quad o_t = S_t q_t = \sum_{\tau=1}^t v_\tau \underbrace{\left\{ k_\tau^\top \left( \prod_{\kappa=\tau+1}^t A_\kappa \right) q_t \right\}}_{\text{Att}_{t,\tau}}, \quad (3)$$

which is either diagonal (Yang et al., 2024a; Gu & Dao, 2023) or scalar-valued (Dao & Gu, 2024). In both, the channels of the hidden state evolve independently. In Equation (3), $\text{Att}_{t,\tau}$ denotes the attention score between $q_t$ and $k_\tau$, while the factor $\prod_{\kappa=\tau+1}^t A_\kappa$ attenuates this inner product according to the cumulative gate between positions $\tau$ and $t$. This term can therefore be interpreted as a position encoding (Yang et al., 2025b), since it depends on the relative distance between $t$ and $\tau$. For Equation (3) to be stable, the gates must be contractive, which is typically enforced by bounding the spectral norm of $A_t$ (Gu & Dao, 2023). More recently, the simple forget gate has been generalized to more expressive *state transition* matrices that also mix channels across time. These are often parameterized in diagonal-plus-low-rank (DPLR) form (Yang et al., 2025a; Peng et al., 2025), which admits a memory-efficient representation of products of such matrices.

***RoPE* and Complex Linear Attention.** Rotary Position Embeddings (*RoPE*) are used to add relative positional information through rotations of the query-key pairs (Su et al., 2021). For queries and keys $q_t, k_\tau \in \mathbb{R}^2$, *RoPE* applies relative positional encoding using the rotation matrix $R_\omega$:

$$\text{Att}_{t,\tau} = \exp\left(k_\tau^\top R_\omega^{t-\tau} q_t\right) = \exp\left((R_\omega^\tau k_\tau)^\top (R_\omega^t q_t)\right), \quad R_\omega = \begin{bmatrix} \cos\omega & -\sin\omega \\ \sin\omega & \cos\omega \end{bmatrix}, \quad (4)$$

with $\omega$ being the frequency of rotation. The query at time $t$ and key at time $\tau$ are rotated via $(R_\omega)^t = R_{t\omega}$. For $d$-dimensional queries and keys, $q_t, k_\tau$ are split into $d/2$ vectors $\in \mathbb{R}^2$, each rotated independently with their own frequency. This yields a block-diagonal rotation matrix $R \in \mathbb{R}^{d \times d}$ where each $2 \times 2$ real matrix $R_{\omega_k}$ is parameterized by a frequency $\omega_k$.

The **RoPE trick** allows expressing a *complex parametrization of a linear Transformer* in the real domain. Consider the real part of the following complex attention score:

$$\text{Att}_{t,\tau} = \Re\{ \tilde{k}_\tau^H \underbrace{\text{diag}\left( \begin{bmatrix} e^{i\omega_1(t-\tau)} & \cdots & e^{i\omega_n(t-\tau)} \end{bmatrix} \right)}_{\bar{R} \in \mathbb{C}^{d/2 \times d/2}} \tilde{q}_t \}, \quad \text{with } \tilde{q}_t, \tilde{k}_\tau \in \mathbb{C}^{d/2}, \quad (5)$$

where $\bar{R}$ is a unitary diagonal state transition matrix. This expression can be understood as applying RoPE to queries and keys $q_t, k_\tau$ in $\mathbb{R}^d$, where we interleave the real and imaginary part in the odd and even indices of queries and keys:

$$\text{Att}_{t,\tau} = \sum_{n=1}^{d/2} \begin{bmatrix} k_{\tau,2n-1} \\ k_{\tau,2n} \end{bmatrix}^\top \underbrace{\begin{bmatrix} \cos\omega_n(t-\tau) & -\sin\omega_n(t-\tau) \\ \sin\omega_n(t-\tau) & \cos\omega_n(t-\tau) \end{bmatrix}}_{R_{\omega_n}^{t-\tau}} \begin{bmatrix} q_{t,2n-1} \\ q_{t,2n} \end{bmatrix}. \quad (6)$$

When we unroll the recurrence in Equation (3) and replace the forget gate, $\boldsymbol{A}_\kappa$, with the block-diagonal rotation matrix $\boldsymbol{R} \in \mathbb{R}^{d \times d}$ in RoPE, we get:

$$\boldsymbol{o}_t = \sum_{\tau=1}^t \boldsymbol{v}_\tau \left\{ \boldsymbol{k}_\tau^\top \boldsymbol{R}^{t-\tau} \boldsymbol{q}_t \right\} \qquad \text{with } \boldsymbol{R}^{t-\tau} = \text{blockdiag}\left( \begin{bmatrix} \boldsymbol{R}_{\omega_1}^{t-\tau} & \cdots & \boldsymbol{R}_{\omega_n}^{t-\tau} \end{bmatrix} \right) \qquad (7)$$

Note that due to the block-diagonal structure of $\boldsymbol{R}$, we can write $\boldsymbol{R}^{t-\tau} = (\boldsymbol{R}^\tau)^\top \boldsymbol{R}^t$, from which follows that $\boldsymbol{k}_\tau^\top \boldsymbol{R}^{t-\tau} \boldsymbol{q}_t = (\boldsymbol{R}^\tau \boldsymbol{k}_\tau)^\top \boldsymbol{R}^t \boldsymbol{q}_t$. This allows us to express the rotation matrix as applying RoPE to queries and keys, similar to Equation (6).

In summary, a linear Transformer with RoPE is *equivalent to the same model with a unitary, diagonal and non-selective transition in half the dimensions*. The *RoPE trick* allows us to implement this complex parameterization by applying RoPE to queries and keys, effectively staying in the real domain which, allows us to re-use existing (linear) attention kernels. A full derivation is shown in Appendix A.1.

## 3  A UNIFYING VIEW: DECAY AND ROTATION

In this section, we motivate our method, *Selective RoPE*, by first observing that pure softmax attention implicitly performs selective rotations when viewed through the lens of Random Fourier Features (RFFs) (Section 3.1). These rotations are missing in linear attention. In Section 3.2, we explain why rotations do not suffice and why selective gating is necessary, building on the complementary roles that real (gating) and imaginary (rotation) parts play in diagonal SSMs. Finally, in Section 3.3 we combine the previous insights and present our proposed method.

### 3.1  SOFTMAX ATTENTION IMPLICITLY PERFORMS ROTATIONS

We begin with the connection between RFFs and softmax attention, and illustrate that rotation is an integral component in softmax attention. Specifically, we start from the definition of the softmax attention in Equation (1) (omitting temperature for simplicity). Following Peng et al. (2021) and Rahimi & Recht (2007, Theorem 1), we define the RFF kernel as $\phi_\omega(\boldsymbol{x}) = \exp\left( \|\boldsymbol{x}\|_2^2 / 2 + i\boldsymbol{\omega}^\top \boldsymbol{x} \right)$ with $\boldsymbol{\omega} \sim \mathcal{N}(0, \boldsymbol{I})$. When applying the kernel to the dot-product of queries and keys $\langle \boldsymbol{q}_t, \boldsymbol{k}_\tau \rangle$, whose expected real component is equivalent to the attention score $\text{Att}_{t,\tau}$, we obtain:

$$\Re \left\{ \mathbb{E}_{\boldsymbol{\omega} \sim \mathcal{N}(0,\boldsymbol{I})} \left[ \phi_\omega(\boldsymbol{q}_t)^\top \phi_\omega(\boldsymbol{k}_\tau) \right] \right\} = \exp\left( \boldsymbol{q}_t^\top \boldsymbol{k}_\tau \right). \qquad (8)$$

By the law of large numbers, with i.i.d. $\boldsymbol{\omega}_j \sim \mathcal{N}\left(0, \sigma^2 \boldsymbol{I}\right)$ for $j \in \{1, \cdots, D\}$ and $\sigma = 1$ we can approximate the output of softmax attention before normalizing the attention scores as:

$$\boldsymbol{s}_t = \Re \left\{ \mathbb{E} \left[ \sum_{\tau=1}^t \phi_\omega(\boldsymbol{q}_t)^\mathsf{H} \phi_\omega(\boldsymbol{k}_\tau) \cdot \boldsymbol{v}_\tau \right] \right\} = \lim_{D \to \infty} \Re \left\{ \frac{1}{D} \sum_{j=1}^D \hat{\boldsymbol{s}}_{t,j} \right\}$$

where $\hat{\boldsymbol{s}}_{t,j} = \sum_{\tau=1}^t \phi_{\omega_j}(\boldsymbol{q}_t)^\mathsf{H} \phi_{\omega_j}(\boldsymbol{k}_\tau) \cdot \boldsymbol{v}_\tau$ is the $j$-th contribution to the attention score $\text{Att}_{t,\tau}$. With some algebraic manipulations and mild assumptions (full derivation in Appendix A.2) and using the definition of $\phi_{\omega_j}$, we can re-express $\hat{\boldsymbol{s}}_j$ as a recurrence. Stacking $D$ of these recurrences horizontally gives us a matrix-valued recurrence over $\hat{\boldsymbol{S}}_t \in \mathbb{R}^{d \times D}$:

$$\hat{\boldsymbol{S}}_t = \hat{\boldsymbol{S}}_{t-1} \bar{\boldsymbol{R}}_t + \boldsymbol{v}_t \tilde{\boldsymbol{k}}_t^\mathsf{H}, \quad \bar{\boldsymbol{R}}_t = \text{diag}\left( \exp\left( i\boldsymbol{\Omega}(\boldsymbol{q}_t - \boldsymbol{q}_{t-1}) \right) \right), \quad \tilde{\boldsymbol{k}}_t = \phi(\boldsymbol{q}_t) \odot \phi(\boldsymbol{k}_\tau), \qquad (9)$$

Crucially, $\bar{\boldsymbol{R}}_t$ is a diagonal *input-dependent rotation matrix* parametrized by random Gaussian features $\boldsymbol{\Omega}$, conditioned on the input via $\boldsymbol{q}_t - \boldsymbol{q}_{t-1}$. Furthermore, due to the corresponding telescoping product, the induced rotation is path independent. Consequently, the rotation introduced by the softmax attention itself lacks an explicit positional inductive bias or encoding. Therefore, the relevance of this expression is structural, allowing us to introduce an efficient way to compute a general form of $\bar{\boldsymbol{R}}_t$, the significance of which will become clear.

Recalling the RoPE trick in Section 2, it should become clear that we can re-express $\bar{\boldsymbol{R}}_t$ as a block-diagonal matrix where each $2 \times 2$ rotation matrix on its diagonal rotates by the angle $\phi_j = \langle \boldsymbol{\omega}_j, (\boldsymbol{q}_t - \boldsymbol{q}_{t-1}) \rangle$. This is a stable recurrence, since $\boldsymbol{R}_t$ is norm preserving, which also means that it does not forget past key-value associations. Interestingly, the shift over the queries $\boldsymbol{q}$ can be expressed by a 1d short-convolution, which is a component that is already frequently used in recurrent architectures (Yang et al., 2025a; Dao & Gu, 2024). We can follow a similar derivation

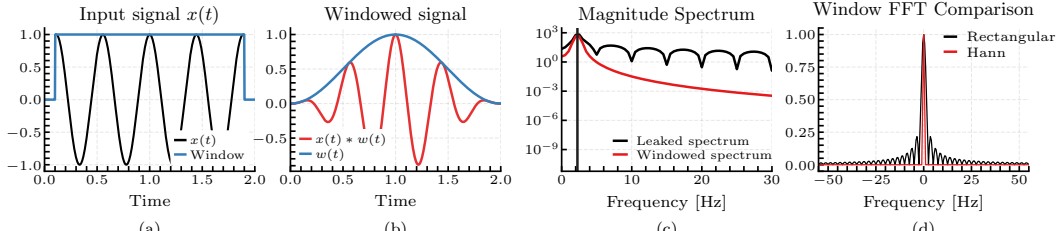

Figure 3: The effects of windowing on the spectrogram of a finite sample of a sequence.

as in Equation (9) for the normalizer $z_t$. The read-out proceeds slightly differently than in normal linear attention: since each column $j$ of the recurrent state represents the contribution of the $j$-th random feature to the approximation of $s_t$, we sum over the columns: $\hat{S}_t \mathbf{1}$.

The equivalence of the RFF kernel in Equation (8) only holds for an unlimited number of samples, i.e., $D \rightarrow \infty$. However, as shown in Theorem 1 (Appendix A.3), in the presence of limited samples, one can have an optimal variance for the RFFs for a single query-key pair, proportional to the angle between the two vectors. Assuming a uniform distribution for the angle between the query-key pairs, we obtain a spectrum of optimal variances.

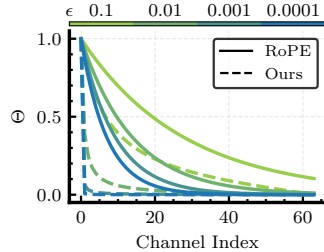

Extending this perspective, we define the rotation matrix as $\hat{R}_t = \exp(i\mathbf{\Omega}\Theta(q_t - q_{t-1}))$, where $\Theta$ is a diagonal matrix of temperatures. Assuming the angle between the queries and keys are uniformly distributed in $[0, 2\pi]$, the optimal temperatures follow $\tan^2(\frac{\theta}{2})$ with $\theta \sim \mathcal{U}[0, 2\pi]$ (ref. Theorem 1). Interestingly, this distribution closely resembles the exponentially decaying frequencies used in *RoPE*, with a slightly faster decline, as shown in Figure 2.

Figure 2: Distribution of phase temperatures in RoPE vs. *Selective RoPE*. $\epsilon$ is the inverse of the RoPE base frequency and the upper-bound of query-key angle in our temperature (details in App. A.3.1).

In summary, we have shown that softmax attention implicitly performs random input-dependent rotations to encode relative positional information between tokens. A comparison to other kernelized linear-attention variants (e.g., Performer, RoFormer, and CosFormer) is provided in Appendix A.6.

## 3.2 NECESSITY OF GATING: SPECTRAL LEAKAGE IN DIAGONAL SSMs

By analyzing the role of real and imaginary parts in complex diagonal SSMs, in this section we show that rotations alone are not enough to close the gap between linear and softmax attention. Inspired by the findings of Section 3.1, we analyze a model that is related to GLA, where the diagonal gate $A_t$ is instead replaced by the rotation matrix $\bar{R}_t$ introduced in Equation (9):

$$S_t = S_{t-1}\bar{R}_t + v_t k_t^{\mathsf{H}}, \quad o_t = \Re\{S_t q_t\}. \qquad (10)$$

By unrolling the recurrence, we can write the output as:

$$o_t = \Re\left\{\sum_{j=1}^{d/2} q_{t,j}\, e^{i\omega_{t,j}} \sum_{\tau=-\infty}^{+\infty} k_{\tau,j}\, e^{-i\omega_{\tau,j}} v_\tau u_t(\tau)d\tau\right\}.$$

This equation contains a convolution of the value and key (i.e., the input) with an exponential function with pure imaginary exponents (i.e., $e^{-i\omega_{\tau,j}}$). This is a spectral analysis (discrete Fourier transform, DFT) of the value signal in the presence of the step-window function $u_t(\tau)$ (definition in Appendix A.4), which is visualized in Figure 3a. When naively performing a DFT over a finite sample, the resulting discontinuities at the margins of the sample cause spectral leakage in the spectrogram as shown in Figure 3c. To avoid spectral leakage, one usually places a non-rectangular window which tapers off towards the margins. The convolution of the signal with a Hann window (Oppenheim, 1999) function is shown in Figure 3b and the resulting magnitude spectrum in Figure 3c. In Figure 3d, we show that we are able to recover the correct frequency after a window FFT when applying a Hann window to our input signal. The window function chosen here acts like an exponential decay towards the margins, which is analogous to using a gate in our model in Equation (10). The use of gates in sequence models has a long history. Starting from the gating mechanism in LSTMs (Hochreiter & Schmidhuber, 1997), it is also widely used in linear attention, linear RNNs and SSMs (Yang et al., 2024a; Gu & Dao, 2023), and even softmax Transformers (Lin et al., 2025). Our results in this section provide a theoretical motivation for the use of gating mechanisms.

### 3.3 Design Principles for Linear Attention

In this section we combine the insights gained in Sections 3.1 and 3.2 to formulate general design principles that are required to narrow the gap between linear and softmax attention. For this, we analyze a general form of linear attention, which encompasses both models in Equation (3) and Equation (10):

$$S_t = S_{t-1} A_t + v_t \tilde{k}_t^{\mathsf{H}}, \quad o_t = \Re\{S_t \tilde{q}_t\}, \quad o_t = \sum_{\tau=1}^{t} v_\tau \Re\left\{ \tilde{k}_\tau^{\mathsf{H}} \left( \prod_{\kappa=\tau}^{t} A_\kappa \right) \tilde{q}_t \right\}. \quad (11)$$

In Section 3.1 we showed that softmax attention implicitly performs input-dependent rotations, a capability that is absent in standard linear attention. We can inject rotations into the general form in eq. (11) by setting $A_t = \bar{R}_t$. Since $\bar{R}_t$ is a rotation matrix, this choice is stable and recovers the model in Equation (10). However, a purely rotational (complex) linear recurrent model behaves like a spectral analyzer: positional information is encoded through phase, but without any decay it cannot effectively control the contribution of distant tokens and suffers from spectral leakage. To remedy this, we also need a decaying (window) component, for which we choose an exponentially decaying gate. Setting $A_t = \Lambda_t$ yields the gated model in Equation (3). In summary, a well-performing linear Transformer requires both (a) *rotation* and (b) *gating*.

Both components can be combined by factorizing the transition as $A_t = \Lambda_t \bar{R}_t$. Interestingly, in DeltaNet the rotational component is already present in the form of an input-dependent Householder transform, which performs rotations along the key dimension. Adding an explicit forget gate on top of this, as done by Yang et al. (2025a), improves performance in line with our design principle. For softmax Transformers, the rotational component is already provided implicitly along random axes (cf. Section 3.1); thus, only a forget gate is needed to fully satisfy the rotation + decay recipe, as empirically validated by the Forgetting Transformer (Lin et al., 2025).

In summary, as the main contribution of the paper, we introduce *Selective RoPE*, which we define as Linear Attention with an input-dependent rotation matrix $R_t$ as its state transition:

$$S_t = S_{t-1} R_t + v_t k_t^\top, \quad o_t = S_t q_t. \quad (12)$$

Recalling the RoPE trick in Equation (7) and defining $R_{i:j} = \prod_{\kappa=i}^{j} R_\kappa$ for the input-dependent rotation matrix $R_\kappa$, we can equivalently write this as:

$$\textbf{\textit{Selective RoPE}:} \quad o_t = \sum_{\tau=1}^{t} v_\tau \left\{ k_\tau^\top R_{\tau+1:t} q_t \right\} = \sum_{\tau=1}^{t} v_\tau \left\{ k_\tau^\top R_{1:\tau}^\top R_{1:t} q_t \right\}. \quad (13)$$

Here, $\hat{R}_t$ should be understood as an input-dependent state transition. Although it is parameterized similar to Equation (9), this does not mean that a query is being written into memory; rather, it should be seen as a selective modulation of the state transition. This can easily be applied to both queries and keys, allowing us to reuse many parts of existing RoPE kernels. Considering the extensive research done on the forget gate, we rely on the built-in forgetting functionality of the baseline architectures.

In summary, in this section we provide theoretical results that motivate the use of complex rotation and exponential decay in a linear attention model. The resulting design principle argues that both these components are required for a well-performing sequence model. This design principle also provides a fresh perspective on the success of variants of softmax Transformers (Press et al., 2021; Lin et al., 2025) and DeltaNet (Yang et al., 2024b; 2025a), which we further elaborate on in Appendix A.7 and Appendix A.5.

## 4 Experiments

In the following section we test our proposed model on synthetic and real-world language modeling tasks. For this we first provide our implementation details and then explain the specific experimental setup for each task and discuss the accompanying results. We primarily apply *Selective RoPE* to Gated Linear Attention (GLA) (Yang et al., 2024a) and compare with other linear and softmax attention variants. We sweep learning rates (reported in Appendix B) unless otherwise specified.

### 4.1 Implementation

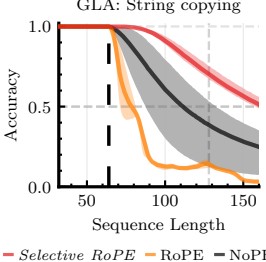

Figure 5: Copying accuracy of GLA with CIs. Dashed line is the training sequence length.

Table 1: MAD benchmark results. We ablate the effectiveness of each extra component introduced to *Selective RoPE* on GLA. The best results are marked in **bold** and the second best in underline.

| Model | Compress | Fuzzy Recall | In-Context Recall | Memorize | Noisy Recall | Selective Copy | Average |
|---|---|---|---|---|---|---|---|
| GLA | | | | | | | |
| *NoPE* | 82.0 | 8.5 | 87.3 | 38.7 | 87.6 | 91.1 | 65.9 |
| *RoPE* | 85.2 | 7.5 | 92.6 | **61.4** | 91.9 | **96.4** | 72.5 |
| *Selective RoPE* | 85.2 | **9.0** | 94.0 | 57.1 | 91.7 | 94.9 | 72.0 |
| + phase gate | 85.1 | 7.5 | **96.6** | 56.9 | 94.3 | 93.5 | 72.3 |
| + bias | 85.0 | 8.4 | 95.0 | 61.3 | 91.2 | 95.4 | 72.7 |
| + phase gate & bias | **85.4** | 7.2 | 95.9 | 60.4 | **95.0** | 95.6 | **73.2** |

In the implementation of *Selective RoPE* we make several design choices that go beyond the architecture described in Section 3.3: Following Zhang et al. (2024), where learning the random features introduced by Choromanski et al. (2021a) was shown to be more effective, we make the parameters $\omega$ in *Selective RoPE* learnable. This makes the rotations input-dependent and learnable. Following Yang et al. (2025b), we place a sigmoid gate on the rotation angles to allow the model to control whether to rotate

```
def s_rope(q, k, W_omega, temp):
    omega = W_omega@q
    omega = conv1d(omega)
    omega = temp*cumsum(omega)
    sin_o, cos_o = sincos(omega)
    return rope(q, k, cos_o, sin_o)
```

Figure 4: Pseudocode of *Selective RoPE*.

or not. We also add a learnable bias term, which is not dependent on relative token positions (Li et al., 2024). Finally, we place a weight norm (Kingma, 2016) on the input projection. We ablate our architectural choices on the MAD dataset and language modeling experiments.

We implement *Selective RoPE* in PyTorch and integrate it into `flash-linear-attention` (Yang & Zhang, 2024) for our experiments. Using the RoPE trick, we are able to implement our method as a prelude to RoPE where we determine the sin and cos from the input as shown in Figure 4. To optimize the throughput of our implementation, we follow the GPT-NeoX (Black et al., 2022) style of applying rotations to allow for coalesced memory access. This is equivalent to our derivations up to an index permutation.

## 4.2 SYNTHETIC LANGUAGE TASKS

To investigate which capabilities of linear attention are improved when using *Selective RoPE*, we run experiments on synthetic tasks. For this, we mostly focus on recall, since it is essential for language modeling (Arora et al., 2024a;b) and a good proxy for performance at scale.

**MQAR.** We evaluate GLA + *Selective RoPE* on Multi-Query Associative Recall, following the same experimental setup as in Arora et al. (2024a, Figure 2) with a finer learning rate grid, as this has been shown to improve performance (Okpekpe & Orvieto, 2025) (cf. Appendix B.2). The results in Figure 6 show that GLA improves with extra positional information and that *Selective RoPE* achieves the greatest improvement over the base model with no positional embedding.

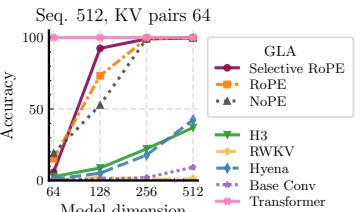

Figure 6: MQAR results.

**MAD and Copying.** We also evaluate our method on the MAD benchmark suite (Poli et al., 2024) which tests a model's ability to store and recall information within its context. Across MAD, *Selective RoPE* improves over NoPE and matches or improves upon RoPE on several recall-oriented tasks; the phase-gated variant achieves the strongest overall average. This task differs from *Selective Copy* in MAD in that the entire input sequence has to be copied token-by-token after the model is presented with a `<copy>` token. The results in Figure 5 show that *Selective RoPE* again improves over the alternatives and learns to length extrapolate very robustly. The poor result of RoPE is reported in prior works (Jelassi et al., 2024; Li et al., 2024) and attributable to its generally poor length extrapolation performance without fine-tuning on longer sequence lengths.

**State Tracking.** A common way to evaluate the expressivity of a model is *state tracking* on permutation composition (Liu et al., 2023). Recently, it has been shown that SSMs and linear RNNs are not capable of learning parity (Merrill et al., 2024), which amounts to permutation composition

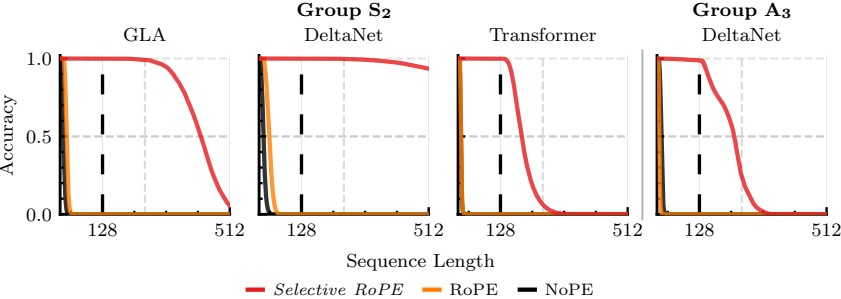

Figure 7: State tracking peformance of GLA, Transformer, and DeltaNet with different positional embeddings on $S_2$ and $A_3$. The models on $S_2$ were trained with *one* layer whereas DeltaNet was trained with *two* layers on $A_3$. Vertical dashed line indicates training sequence length.

on the symmetric group of two elements, $S_2$, and that one needs to extend the eigenvalue range of the state transition $\boldsymbol{A}_t$ from $[0, 1]$ to $[-1, 1]$ (Grazzi et al., 2025). In Figure 7 we see that GLA with *Selective RoPE* is able to learn and length-extrapolate on $S_2$. This is in line with our expectations since the input dependent rotations allow it to model "flips" depending on the input either being a 0 or a 1, while GLA with NoPE and RoPE does not even learn the training context length. Similarly, we can see that *Selective RoPE* also improves the state tracking abilities in Transformers (i.e., softmax attention) allowing them to solve the parity problem up to, and slightly more, than the train sequence length. To the best of our knowledge, Transformer with *Selective RoPE* is the only variant of Transformers capable of solving the parity task with a single layer up to this sequence length (Liu et al., 2023). We also experiment on $A_3$ with a 2-layer DeltaNet (Yang et al., 2024b), which is the permutation composition on the symmetric group of three elements, limited to even permutations. As we can observe, *Selective RoPE* improves the expressivity of the model up to a point where it is capable of solving $A_3$ up to the training sequence length. To the best of our knowledge, this is the first time these results have been presented for our choice of model on this task.

### 4.3 LANGUAGE MODELING

For our language modeling experiments we train 370M and 1.3B parameter versions of GLA (Yang et al., 2024a) and the Forgetting Transformer (FoX) (Lin et al., 2025) using AdamW (Loshchilov & Hutter, 2019) and a warmup and cosine-decay schedule (Loshchilov & Hutter, 2017). All models are trained on 10B and 26B tokens respectively. We use FineWeb (Penedo et al., 2024) at a context length of 4096 and use the Mistral 7B tokenizer (Jiang et al., 2023) with a vocabulary size of 32 000. All remaining architectural and optimizer hyperparameters (batch size, learning rate schedule, gradient clipping, weight decay) follow Siems et al. (2025) and are detailed in Appendix B. To account for differences in optimal learning rates for the considered positional embedding schemes, we sweep learning rates following Orvieto & Gower (2025). To select the best learning rate for each model and position embedding combination, we use the perplexity on 4 million tokens not seen during training. The best models are then evaluated on downstream tasks from `lm-eval-harness` (Gao et al., 2024), the results of which are shown in Table 3. We follow the default zero-shot evaluation setup in `lm-eval-harness`, using its standard prompting and report the macro-average accuracy over the core multiple-choice tasks in the Avg. column. We select the same set of tasks as in GLA (Yang et al., 2024a).

Initial experiments have shown that Selective RoPE displays training instabilities at higher learning rates, earlier than models trained with RoPE or NoPE. We find that this can be remedied by using the $\ell^2$-normalized input, $\boldsymbol{x}_t$, instead of the queries, $\boldsymbol{q}_t$, to parametrize the rotation angles $\omega$. Further stability improvements were achieved by either placing a weight norm (Kingma, 2016) on the $\omega$ projection or using a low-rank MLP with rank 8 or 16. We also test adding a learnable bias term (Li et al., 2024) but find that this does not significantly improve performance. We find that the most significant performance improvements come from adding a SiLU activation after the

Table 2: Selective RoPE ablations. GLA 370M trained on 10B tokens of FineWeb.

| Model | Avg-PPL ↓ | Avg-Acc ↑ |
|---|---|---|
| NoPE | **20.78** | 46.8 |
| RoPE | 21.42 | 46.5 |
| *Selective RoPE* | | |
|   Baseline | 21.27 | 45.8 |
|   + SiLU | 21.14 | 46.7 |
|   + phase gate | 21.16 | **47.1** |

| Model | Wiki. ppl ↓ | LMB. ppl ↓ | LMB. acc ↑ | PIQA acc ↑ | Hella. acc_n ↑ | Wino. acc ↑ | ARC-e acc ↑ | ARC-c acc_n ↑ | Avg. |
|---|---|---|---|---|---|---|---|---|---|
| *FoX 370M params / 10B tokens* | | | | | | | | | |
| NoPE | 25.52 | 17.76 | 43.0 | 67.0 | 42.2 | 52.8 | 47.1 | 25.0 | 45.3 |
| RoPE | **25.29** | 17.71 | 42.2 | 68.4 | 41.9 | 50.8 | 47.7 | 24.6 | 45.1 |
| Selective RoPE | 33.87 | **17.29** | 43.5 | 68.7 | 42.5 | 53.0 | 48.4 | 25.7 | 46.1 |
| *GLA 370M params / 10B tokens* | | | | | | | | | |
| NoPE | 25.72 | **15.83** | 43.1 | 67.8 | 43.1 | 52.2 | 49.4 | 24.9 | 46.8 |
| RoPE | 26.51 | 16.32 | 42.5 | 69.6 | 43.3 | 50.7 | 48.2 | 24.5 | 46.5 |
| Selective RoPE | 25.98 | 16.33 | 42.2 | 69.1 | 43.8 | 51.3 | 49.9 | 26.6 | 47.1 |
| *Gated DeltaNet ∼400M params / 10B tokens* | | | | | | | | | |
| NoPE | 25.82 | 16.27 | 43.0 | 68.6 | 43.3 | 52.2 | 48.4 | 24.2 | 46.6 |
| RoPE | 25.92 | 16.79 | 41.7 | 69.5 | 43.3 | 52.5 | 49.2 | 24.1 | 46.7 |
| Selective RoPE[†] | 26.13 | **15.93** | 42.3 | 69.1 | 44.3 | 54.1 | 49.9 | 25.5 | 47.5 |
| *GLA 1.3B params / 26B tokens* | | | | | | | | | |
| NoPE | 18.18 | 8.59 | 53.8 | 73.3 | 56.5 | 58.2 | 59.0 | 30.0 | 55.2 |
| RoPE | 18.50 | 8.88 | 53.0 | 72.4 | 56.6 | 57.1 | 58.5 | 28.5 | 54.4 |
| Selective RoPE | **17.87** | **8.50** | 53.8 | 73.1 | 56.9 | 56.0 | 59.3 | 28.8 | 54.6 |

Table 3: Evaluation results on tasks from `lm-eval-harness` (Gao et al., 2024). FoX rows correspond to the output-norm-OFF setting only. GLA blocks are kept from the original table. Gated DeltaNet rows are added from the local ∼400M-parameter / 10B-token sweep, selecting the best checkpoint by final validation perplexity within each positional-encoding setting. For Gated DeltaNet, the selective row ([†]) corresponds to the available `stable_selective_rope` variant.

convolution that follows the $\omega$ projection and from adding a gate on the rotation (phase) Yang et al. (2025b) as described in Table 2.

The results in Table 3 use the best configuration of Selective RoPE for both FoX and GLA at 370M and 370 and 1.3B params with 10B and 26B tokens, respectively. We find that Selective RoPE changes the trade-off between perplexity and downstream accuracy in a model-dependent way: it often improves perplexity and/or specific downstream tasks, while the macro-average accuracy may stay similar or decrease slightly depending on architecture and scale.

# 5 RELATED WORK

There have been several attempts at reducing the quadratic complexity of softmax attention (Dao, 2024), one of which is linearization (Katharopoulos et al., 2020), which results in a recurrent model with sub-quadratic cost (Martin & Cundy, 2018; Gu et al., 2020). However, the reduced complexity comes at the cost of lower performance, especially in recall-intensive tasks (Waleffe et al., 2024; Peng et al., 2021; Choromanski et al., 2021a; Zhang et al., 2024). This led to the development of architectures which used gating to increase their expressivity. Non-selective state-space models (SSMs) made use of input-independent gating mechanisms and vector-valued states to perform sequence modeling (Orvieto et al., 2023a; Gu et al., 2022b;a; Sun et al., 2023a). Later, these architectures were improved by adding selective gating (De et al., 2024; Qin et al., 2023) and matrix-valued states (Gu & Dao, 2023; Dao & Gu, 2024; Yang et al., 2024a; Beck et al., 2024; Qin et al., 2024). Concurrently, DeltaNet (Schlag et al., 2021; Yang et al., 2024b) extended the notion of a gate to a state transition matrix by using an input-dependent generalized Householder matrix, which implements the error-correcting delta-rule (Widrow et al., 1988). A byproduct of our theoretical analysis are further insights into the functionality of the gating mechanism and forget gate in Section 3. Another line of work has improved sub-quadratic sequence models through better kernel approximations of softmax attention (Katharopoulos et al., 2020). This approach led to the use of random features (Choromanski et al., 2021a; 2022), which was extended to learning the features directly (Zhang et al., 2024). Interestingly, a polynomial kernel inspired by the Taylor expansion of the exponential function has proved effective in closing the performance gap, while being less efficient in terms of computational complexity (Zhang et al., 2024; Kacham et al., 2023). We base our theoretical investigation on the work of Peng et al. (2021), deriving a linear attention variant as an approximation of the softmax Transformer.

**RoPE and complex parameterizations of RNNs.** The primary method of encoding positional information in sub-quadratic attention variants is exponential decay (Lin et al., 2025). However, in softmax Transformers, rotary position embeddings (RoPE) have proven to be very effective (Su et al., 2021; Shaw et al., 2018; Yang et al., 2025b) compared to no positional embeddings (NoPE) (Kazemnejad et al., 2023). RoPE encodes positional information through point-wise rotation of the query-key pairs. Other variants of RoPE have made attempts at improving RoPE in terms of its shortcomings in generalizing to longer sequences by learning the position embedding (Li et al., 2024), framing it as a kernel design problem (Chi et al., 2022), or utilizing theoretical tools (Peng et al., 2024). Interestingly, our model generalizes RoPE by making angles input-dependent. In our experiments, we show the effectiveness of our proposed position embedding both in linear attention models and softmax Transformers. As shown in Section 2, applying RoPE to a linear Transformer is equivalent to operating in the complex domain and theoretically, this is essential for the universality guarantees of RNNs and SSMs (Orvieto et al., 2024; Gu et al., 2020). Further investigation showed an improvement in the recall capabilities and expressivity of SSMs when operating in the complex domain (Ran-Milo et al., 2024). However, later variants of these models removed the complex recurrence due to inconclusive evidence for their benefits in language modeling and implementation overhead (Gu & Dao, 2023; Dao & Gu, 2024; De et al., 2024). In this paper, we focus on the kernel view of softmax attention, providing a connection between it and linear attention models operating in the complex domain. The resulting design principle provides a connection between softmax attention, complex linear attention, the gating mechanism, and position embeddings.

## 6 CONCLUSION

We introduced *Selective RoPE*, an input-dependent rotary position embedding that generalizes RoPE from fixed to arbitrary, learnable rotations. Our theory shows that (i) softmax attention admits a complex linear formulation that implicitly performs *selective rotations*, and (ii) this complex formulation introduces spectral leakage, which can be suppressed through the forget gate mechanism. Empirically, equipping certain sequence models (namely, GLA, Gated DeltaNet, and softmax Transformers) with *Selective RoPE* improves recall-centric synthetic tasks and strengthens language modeling downstream performance for linear attention variants. Furthermore, we show that this improvement in performance comes at very little computational cost, with an easy implementation thanks to the RoPE trick.

**Future work.** There are several aspects of *Selective RoPE* and the proposed design principle introduced in our paper that require further investigation. Firstly, we note that incorporating RoPE is notoriously detrimental to the length-extrapolation capabilities of sequence models (Li et al., 2024). In this paper, we do not investigate this aspect since we consider it to be out of the scope of our research. Secondly, we believe that further investigation of the effect of the extra components used in *Selective RoPE*, namely the bias term and the phase gate, can be a fruitful direction for future research. Thirdly, we consider the impact of choosing a diagonal as opposed to a scalar forget gate to be an interesting question, since our theoretical justification for forget gates is only concerned with an exponentially decaying component in the sequence model, and not the dimensionality of it. Finally, given the existing variants of RoPE (Black et al., 2022; Su et al., 2021), we believe it to be important to also incorporate the progress on the positional embedding front into future work.

## ACKNOWLEDGEMENTS

We would like to thank Philipp Nazari, Julie Naegelen, Felix Sarnthein, and Gwendolyn Neitzel for constructive discussions and feedback. We would also like to thank Songlin Yang for pointing out an error in a previous version of our manuscript which overstated the expressivity of *Selective RoPE*. This research was partially supported by the following sources: PNRR MUR Project PE000013 CUP J53C22003010006 Future Artificial Intelligence Research (FAIR), funded by the European Union NextGenerationEU, and EU Project ELSA under grant agreement No. 101070617. TAILOR, a project funded by EU Horizon 2020 research and innovation programme under GA No 952215; the Deutsche Forschungsgemeinschaft (DFG, German Research Foundation) under grant number 417962828; the European Research Council (ERC) Consolidator Grant 'Deep Learning 2.0' (grant no. 10). This research was partially funded by the Deutsche Forschungsgemeinschaft (DFG, German Research Foundation) under grant number 539134284, through EFRE (FEIH 2698644) and the state of Baden-Württemberg. Frank Hutter, Antonio Orvieto, Sajad Movahedi, and Timur Carstensen acknowledge financial support by the Hector Foundation. The authors acknowledge support from ELLIS and ELIZA, funded by the European Union. The authors gratefully acknowledge the computing time made available to them on the high-performance computers and at the NHR Centers at TU Dresden and KIT. These centers are jointly supported by the Federal Ministry of Research, Technology and Space of Germany and the state governments participating in the NHR. Arshia Afzal acknowledges support under project ID #37 as part of the Swiss AI Initiative, through a grant from the ETH Domain, and computational resources provided by the Swiss National Supercomputing Centre (CSCS) under the Alps infrastructure. This work was funded by the Swiss National Science Foundation (SNSF) under grant number 2000-1-240094. Research was sponsored by the Army Research Office and was accomplished under Grant Number W911NF-24-1-0048. Views and opinions expressed are however those of the author(s) only and do not necessarily reflect those of the European Union or the ERC. Neither the European Union nor the ERC can be held responsible for them. Finally, we gratefully acknowledge the support of the Schmidt Sciences AI2050 fellowship.

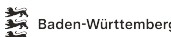 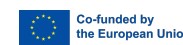

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

The supplementary is structured as follows:

Appendix A contains all derivations and proofs:

- A.1 shows that parameterizing a linear transformer with a unitary diagonal state transition can be implemented by applying RoPE to the queries and keys of the same models.

- A.2 shows that one can use Random Fourier Features (RFFs) to approximate the exponential kernel and thereby softmax attention and, when limiting the approximation to the $D$-dimensions, can be expressed as a recurrent model that can be implemented using an input-dependent variant of RoPE.

- A.3 derives the optimal variance for the RFFs used in Appendix A.2.

- A.4 shows that complex diagonal SSMs can be understood as spectral analyzers that suffer from spectral leakage. A well known remedy for spectral leakage is using real-valued decaying window functions, which can also be seen as forget gates, a prevalent component in sequence models. This highlights the complementary roles of both imaginary and real parts of a gate in recurrent sequence models, with the former rotating and the latter decaying the past observation.

- Appendix A.5 derives the connection between rotation using RoPE and Householder products used in DeltaNet.

Appendix B lists the experimental details for language modeling and synthetic tasks and includes a code listing of the implementation of *Selective RoPE*.

**Notation.** We use the following notation for mathematical objects: Lower-case letters denote scalars $(\alpha, \beta)$. Upper-case bold letters denote matrices $(\boldsymbol{W}, \boldsymbol{A})$. Lower-case bold letters denote vectors $(\boldsymbol{v}, \boldsymbol{k}, \boldsymbol{q})$. $\top$ denotes the transpose operator. H denotes the conjugate transpose operator. $\odot$ denotes the Hadamard-product. Taking the real or imaginary component of an expression is denoted by either $\Re$ or $\Im$. Expressing a vector as a diagonal matrix is denoted by $\mathrm{diag}(\cdot)$. Block-diagonalizing a set of square matrices is denoted by $\mathrm{blockdiag}(\cdot)$. Concatenating vectors is denoted by $\boldsymbol{x}_t = \mathrm{concat}\left([\cdots]^\top\right)$. By $\varphi$ we denote the argument of a complex number.

## A MATHEMATICAL DERIVATIONS AND PROOFS

### A.1 *RoPE* AS IMAGINARY-VALUED LINEAR TRANSFORMER

We start by unrolling the linear Transformers recurrence:

$$\boldsymbol{S}_t = \boldsymbol{S}_{t-1}\bar{\boldsymbol{R}} + \boldsymbol{v}_t\tilde{\boldsymbol{k}}_t^{\mathsf{H}}, \quad \boldsymbol{o}_t = \Re\{\boldsymbol{S}_t\tilde{\boldsymbol{q}}_t\}$$

$$\boldsymbol{o}_t = \Re\left\{\sum_{\tau=1}^{t}\boldsymbol{v}_\tau\tilde{\boldsymbol{k}}_\tau^{\mathsf{H}}\bar{\boldsymbol{R}}^{t-\tau}\tilde{\boldsymbol{q}}_t\right\} = \sum_{\tau=1}^{t}\boldsymbol{v}_\tau\Re\left\{\tilde{\boldsymbol{k}}_\tau^{\mathsf{H}}\bar{\boldsymbol{R}}^{t-\tau}\tilde{\boldsymbol{q}}_t\right\}$$

Therefore, the attention score applied to value $\boldsymbol{v}_\tau$ is:

$$\mathrm{Att}_{t,\tau} = \Re\left\{\tilde{\boldsymbol{k}}_\tau^{\top}\bar{\boldsymbol{R}}^{t-\tau}\tilde{\boldsymbol{q}}_t\right\}$$

Since $\bar{\mathbf{R}}$ is diagonal, we can expand the expression as:

$$\mathbf{Att}_{t\tau} = \Re\left\{\sum_{n=1}^{d/2}(\tilde{\boldsymbol{q}}_{t,n}^R + i\,\tilde{\boldsymbol{q}}_{t,n}^I)\cdot e^{i\omega_n(t-\tau)}\cdot(\tilde{\boldsymbol{k}}_{\tau,n}^R + i\,\tilde{\boldsymbol{k}}_{\tau,n}^I)\right\}$$

$$= \Re\left\{\sum_{n=1}^{d/2}|\tilde{\boldsymbol{q}}_{t,n}|\,e^{-i\varphi(\tilde{\boldsymbol{q}}_{t,n})}\cdot e^{i\omega_n(t-\tau)}\cdot|\tilde{\boldsymbol{k}}_{\tau,n}|\,e^{-i\varphi(\tilde{\boldsymbol{k}}_{\tau,n})}\right\}$$

$$= \Re\left\{\sum_{n=1}^{d/2}|\tilde{\boldsymbol{q}}_{t,n}|\,|\tilde{\boldsymbol{k}}_{\tau,n}|\,e^{i\left(\omega_n(t-\tau)-\varphi(\tilde{\boldsymbol{q}}_{t,n})-\varphi(\tilde{\boldsymbol{k}}_{\tau,n})\right)}\right\}$$

$$= \sum_{n=1}^{d/2}|\tilde{\boldsymbol{q}}_{t,n}|\,|\tilde{\boldsymbol{k}}_{\tau,n}|\cos\left(\omega_n(t-\tau)-\varphi(\tilde{\boldsymbol{q}}_{t,n})-\varphi(\tilde{\boldsymbol{k}}_{\tau,n})\right) \tag{14}$$

where $\varphi(\tilde{\boldsymbol{q}}_{t,n})$ and $\varphi(\tilde{\boldsymbol{k}}_{\tau,n})$ denote the complex phases (angles) of the $n$-th component of $\tilde{\boldsymbol{q}}_t$ and $\tilde{\boldsymbol{k}}_\tau$, respectively. Equation (14) shows that an imaginary forget gate rotates the query-key pairs at each index $n$ with a distinct frequency $\omega_n$. We now demonstrate that this is equivalent to applying *RoPE*. Replacing the cosine in eq. (14) with its matrix multiplication equivalent:

$$\cos\left(\omega_n(t-\tau)-\angle\tilde{\boldsymbol{q}}_{t,n}-\angle\tilde{\boldsymbol{k}}_{\tau,n}\right) = \begin{bmatrix}\cos(\angle\tilde{\boldsymbol{q}}_{t,n})\\\sin(\angle\tilde{\boldsymbol{q}}_{t,n})\end{bmatrix}^\top\begin{bmatrix}\cos(\omega_n(t-\tau)) & -\sin(\omega_n(t-\tau))\\\sin(\omega_n(t-\tau)) & \cos(\omega_n(t-\tau))\end{bmatrix}\begin{bmatrix}\cos(\angle\tilde{\boldsymbol{k}}_{\tau,n})\\\sin(\angle\tilde{\boldsymbol{k}}_{\tau,n})\end{bmatrix}$$

Plugging above in eq. (14) we achieve:

$$\mathrm{Att}_{t,\tau} = \sum_{n=1}^{d/2}|\tilde{\boldsymbol{q}}_{t,n}|\,|\tilde{\boldsymbol{k}}_{\tau,n}|\begin{bmatrix}\cos(\angle\tilde{\boldsymbol{q}}_{t,n})\\\sin(\angle\tilde{\boldsymbol{q}}_{t,n})\end{bmatrix}^\top\begin{bmatrix}\cos(\omega_n(t-\tau)) & -\sin(\omega_n(t-\tau))\\\sin(\omega_n(t-\tau)) & \cos(\omega_n(t-\tau))\end{bmatrix}\begin{bmatrix}\cos(\angle\tilde{\boldsymbol{k}}_{\tau,n})\\\sin(\angle\tilde{\boldsymbol{k}}_{\tau,n})\end{bmatrix}$$

$$= \sum_{n=1}^{d/2}|\tilde{\boldsymbol{q}}_{t,n}|\begin{bmatrix}\cos(\angle\tilde{\boldsymbol{q}}_{t,n})\\\sin(\angle\tilde{\boldsymbol{q}}_{t,n})\end{bmatrix}^\top\begin{bmatrix}\cos(\omega_n(t-\tau)) & -\sin(\omega_n(t-\tau))\\\sin(\omega_n(t-\tau)) & \cos(\omega_n(t-\tau))\end{bmatrix}|\tilde{\boldsymbol{k}}_{\tau,n}|\begin{bmatrix}\cos(\angle\tilde{\boldsymbol{k}}_{\tau,n})\\\sin(\angle\tilde{\boldsymbol{k}}_{\tau,n})\end{bmatrix}$$

$$= \sum_{n=1}^{d/2}\begin{bmatrix}\tilde{\boldsymbol{q}}_{t,n}^R\\\tilde{\boldsymbol{q}}_{t,n}^I\end{bmatrix}^\top\begin{bmatrix}\cos(\omega_n(t-\tau)) & -\sin(\omega_n(t-\tau))\\\sin(\omega_n(t-\tau)) & \cos(\omega_n(t-\tau))\end{bmatrix}\begin{bmatrix}\tilde{\boldsymbol{k}}_{\tau,n}^R\\\tilde{\boldsymbol{k}}_{\tau,n}^I\end{bmatrix} \tag{15}$$

Using the definition of:

$$\boldsymbol{q}_t = \bigoplus_{n=1}^{d/2}\begin{bmatrix}\tilde{\boldsymbol{q}}_{t,n}^R\\\tilde{\boldsymbol{q}}_{t,n}^I\end{bmatrix}, \quad \boldsymbol{k}_\tau = \bigoplus_{n=1}^{d/2}\begin{bmatrix}\tilde{\boldsymbol{k}}_{\tau,n}^R\\\tilde{\boldsymbol{k}}_{\tau,n}^I\end{bmatrix}.$$

we can write Equation (15) as:

$$\mathrm{Att}_{t,\tau} = \sum_{n=1}^{d/2}\boldsymbol{q}_{t,n}\boldsymbol{R}_{\omega_n}^{t-\tau}\boldsymbol{k}_{\tau,n}$$

which is theoretically equivalent to applying *RoPE* to query-key pairs $\boldsymbol{q}_t, \boldsymbol{k}_\tau$. *RoPE* interleaves the real and imaginary parts of complex queries and keys across the hidden dimension, then applies 2D rotations to each pair.

## A.2 Random Fourier Feature Approximation of Softmax Attention

We start with the definition of softmax attention:

$$\boldsymbol{o}_t = \frac{\boldsymbol{s}_t}{\boldsymbol{z}_t}, \quad \boldsymbol{s}_t = \sum_{\tau=1}^{t}\exp\left(\tfrac{1}{\sqrt{d}}\boldsymbol{q}_t^\top\boldsymbol{k}_\tau\right)\cdot\boldsymbol{v}_\tau, \quad \boldsymbol{z}_t = \sum_{\tau=1}^{t}\exp\left(\tfrac{1}{\sqrt{d}}\boldsymbol{q}_t^\top\boldsymbol{k}_\tau\right),$$

where $\boldsymbol{q}_t, \boldsymbol{k}_\tau \in \mathbb{R}^d$. For simplicity, we omit the normalization factor $1/\sqrt{d}$ and first focus on the numerator of the output, specifically the exponential kernel. As in Equation (2), the denominator scaling can be handled separately through an external state $\boldsymbol{z}_t$.

To approximate the exponential kernel $\exp(\cdot)$, we use Random Fourier Features (RFF) (Rahimi & Recht, 2007) with frequencies $\boldsymbol{\omega} \in \mathbb{R}^d \sim \mathcal{N}(0, \sigma^2 \boldsymbol{I})$. The feature map is defined as

$$\phi_{\boldsymbol{\omega}}(\boldsymbol{x}) = \exp\left(\frac{\|\boldsymbol{x}\|_2^2}{2} + i\boldsymbol{\omega}^\top \boldsymbol{x}\right),$$

so that

$$\exp(\boldsymbol{q}_t^\top \boldsymbol{k}_\tau) = \Re\left\{\mathbb{E}_{\boldsymbol{\omega} \sim \mathcal{N}(0,\sigma^2 I)}\left[\phi_{\boldsymbol{\omega}}(\boldsymbol{q}_t)^\top \phi_{\boldsymbol{\omega}}(\boldsymbol{k}_\tau)\right]\right\},$$

for $\sigma = 1$. By applying this feature map to the linear attention formulation in Equation (2), we can approximate the exponential kernel in softmax attention. Continuing the approximation:

$$\exp(\boldsymbol{q}_t^\top \boldsymbol{k}_\tau) = \exp\left(\frac{\|\boldsymbol{q}_t\|_2^2 + \|\boldsymbol{k}_\tau\|_2^2}{2}\right) \cdot \Re\left\{\mathbb{E}_{\boldsymbol{\omega} \sim \mathcal{N}(0,\boldsymbol{I})}\left[\exp(i\boldsymbol{\omega}^\top \boldsymbol{q}_t) \exp(-i\boldsymbol{\omega}^\top \boldsymbol{k}_\tau)\right]\right\}.$$

Let $\boldsymbol{\omega}_j \sim \mathcal{N}\left(0, \sigma^2 \boldsymbol{I}\right)$ for $j \in \{1, 2, \ldots, D\}$. Then due to the law of large numbers we have:

$$\exp(\boldsymbol{q}_t^\top \boldsymbol{k}_\tau) = \exp\left(\frac{\|\boldsymbol{q}_t\|_2^2 + \|\boldsymbol{k}_\tau\|_2^2}{2}\right) \cdot \Re\left\{\lim_{D \to \infty} \frac{1}{D} \sum_{j=1}^{D} \exp(i\boldsymbol{\omega}_j^\top \boldsymbol{q}_t) \cdot \exp(-i\boldsymbol{\omega}_j^\top \boldsymbol{k}_\tau)\right\}.$$

Therefore, we can approximate $\exp(\boldsymbol{q}_t^\top \boldsymbol{k}_\tau)$ as the dot product of the random exponential projection of the query and the key using $D$ random $\boldsymbol{\omega}_j$s:

$$\hat{\boldsymbol{s}}_t^D = \frac{1}{D} \sum_{\tau=1}^{t} \sum_{j=1}^{D} \exp\left(\frac{\|\boldsymbol{q}_t\|_2^2 + \|\boldsymbol{k}_\tau\|_2^2}{2}\right) \exp(i\boldsymbol{\omega}_j^\top \boldsymbol{q}_t) \exp(-i\boldsymbol{\omega}_j^\top \boldsymbol{k}_\tau) \cdot \boldsymbol{v}_\tau.$$

This allows us to compute the softmax attention as the linear attention parameterized by:

$$\phi(\boldsymbol{q}_t) = \exp\left(\frac{\|\boldsymbol{q}_t\|_2^2}{2}\right) \cdot \exp(i\boldsymbol{\Omega}^\top \boldsymbol{q}_t), \quad \phi(\boldsymbol{k}_\tau) = \exp\left(\frac{\|\boldsymbol{k}_t\|_2^2}{2}\right) \cdot \exp(-i\boldsymbol{\Omega}^\top \boldsymbol{k}_\tau),$$

with $\lim_{D \to \infty} \Re\left\{\hat{\boldsymbol{s}}_t^D\right\} = \sum_{\tau=1}^{t} \exp(\boldsymbol{q}_t^\top \boldsymbol{k}_\tau) \cdot \boldsymbol{v}_\tau$ and $\boldsymbol{\Omega} = [\boldsymbol{\omega}_1, ..., \boldsymbol{\omega}_D]$. Omitting the superscript $D$ for simplifying the notation, let us focus on one random feature $\boldsymbol{\omega}_j$ and its contribution to the output:

$$\hat{\boldsymbol{s}}_{t,j} = \sum_{\tau=1}^{t} \exp\left(\frac{\|\boldsymbol{q}_t\|_2^2}{2}\right) \exp\left(\frac{\|\boldsymbol{k}_\tau\|_2^2}{2}\right) \exp(i\boldsymbol{\omega}_j^\top \boldsymbol{q}_t) \exp(-i\boldsymbol{\omega}_j^\top \boldsymbol{k}_\tau) \cdot \boldsymbol{v}_\tau.$$

In this case, we have $\hat{\boldsymbol{s}}_t^D = \frac{1}{D} \hat{\boldsymbol{S}}_t^D \boldsymbol{1}$, where $\hat{\boldsymbol{S}}_t^D = [\hat{\boldsymbol{s}}_{t,1} \quad \hat{\boldsymbol{s}}_{t,2} \quad \ldots \quad \hat{\boldsymbol{s}}_{t,D}] \in \mathbb{C}^{d \times D}$. Now note that we have:

$$\hat{\boldsymbol{s}}_{t,j} = \sum_{\tau=1}^{t-1} \exp\left(\frac{\|\boldsymbol{q}_t\|_2^2 - \|\boldsymbol{q}_{t-1}\|_2^2}{2}\right) \exp(i\boldsymbol{\omega}_j^\top \boldsymbol{q}_{t-1}) \exp(i\boldsymbol{\omega}_j^\top (\boldsymbol{q}_t - \boldsymbol{q}_{t-1})) \exp(-i\boldsymbol{\omega}_j^\top \boldsymbol{k}_\tau) \cdot \boldsymbol{v}_\tau \quad (16)$$

$$+ \exp\left(\frac{\|\boldsymbol{q}_t\|_2^2}{2}\right) \exp\left(\frac{\|\boldsymbol{k}_t\|_2^2}{2}\right) \exp(i\boldsymbol{\omega}_j^\top (\boldsymbol{q}_t - \boldsymbol{k}_t)) \cdot \boldsymbol{v}_t. \quad (17)$$

$$= \exp\left(\frac{\|\boldsymbol{q}_t\|_2^2 - \|\boldsymbol{q}_{t-1}\|_2^2}{2}\right) \exp(i\boldsymbol{\omega}_j^\top (\boldsymbol{q}_t - \boldsymbol{q}_{t-1})) \hat{\boldsymbol{s}}_{t-1}^j + \phi_{\boldsymbol{\omega}_j}(\boldsymbol{q}_t) \cdot \phi_{\boldsymbol{\omega}_j}(\boldsymbol{k}_t) \cdot \boldsymbol{v}_t \quad (18)$$

Note that the real exponential component in Equation (18) can introduce instability to the recurrence. Therefore, following the standard in both linear Transformers (Yang et al., 2024b;a; 2025a; Lin et al., 2025) and deep softmax Transformers (Henry et al., 2020), we assume $L_2$ normalization over the query and the key, i.e., $\|\boldsymbol{q}_t\|_2 = \|\boldsymbol{q}_{t-1}\|_2$. Thus, recurrence presented in Equation (18) simplifies to:

$$\hat{\boldsymbol{s}}_{t,j} = \exp(i\boldsymbol{\omega}_j^\top (\boldsymbol{q}_t - \boldsymbol{q}_{t-1})) \hat{\boldsymbol{s}}_{t-1,j} + \phi_{\boldsymbol{\omega}_j}(\boldsymbol{q}_t) \cdot \phi_{\boldsymbol{\omega}_j}(\boldsymbol{k}_t) \cdot \boldsymbol{v}_t,$$

with $\hat{\boldsymbol{s}}_{t,j}$ being the $j^{th}$ column of $\hat{\boldsymbol{S}}_t^D$ is scaled by the values $\exp(i\boldsymbol{\omega}_j^\top (\boldsymbol{q}_t - \boldsymbol{q}_{t-1}))$. Therefore, we can write the recurrence over $\hat{\boldsymbol{S}}_t$ as:

$$\hat{\boldsymbol{S}}_t^D = \hat{\boldsymbol{S}}_{t-1} \bar{\boldsymbol{R}}_t + \boldsymbol{v}_t \left(\phi(\boldsymbol{q}_t) \circ \phi(\boldsymbol{k}_t)\right)^\top, \quad \hat{\boldsymbol{s}}_t^D = \frac{1}{D} \hat{\boldsymbol{S}}_t^D \boldsymbol{1}.$$

where $\phi(x)$ is a vector with its $j^{th}$ element equal to $\phi_{\boldsymbol{\omega}_j}(x)$, and $\bar{\boldsymbol{R}}_t$ is:

$$\bar{\boldsymbol{R}}_t = \text{diag}(\exp(i\boldsymbol{\Omega}^\top (\boldsymbol{q}_t - \boldsymbol{q}_{t-1}))) \quad (19)$$

Focusing on Equation (19), we observe that exponential kernel in softmax attention implicitly applies a form of input-dependent *(Selective) RoPE* (see Sec. 2). However, instead of learning the frequencies $\mathbf{\Omega}$, they are randomly sampled from a normal distribution.

Similarly, we can also approximate the normalizing factor $z_t$ as:

$$\hat{z}_t^D = \frac{1}{D} \sum_{\tau=1}^{t} \sum_{j=1}^{D} \exp\left(\frac{\|\boldsymbol{q}_t\|_2^2 + \|\boldsymbol{k}_\tau\|_2^2}{2}\right) \exp\left(i\boldsymbol{\omega}_j^\top \boldsymbol{q}_t\right) \exp\left(-i\boldsymbol{\omega}_j^\top \boldsymbol{k}_\tau\right).$$

Separating the contribution of each random feature, we have:

$$\hat{z}_{t,j} = \sum_{\tau=1}^{t} \exp\left(\frac{\|\boldsymbol{q}_t\|_2^2}{2}\right) \exp\left(\frac{\|\boldsymbol{k}_\tau\|_2^2}{2}\right) \exp\left(i\boldsymbol{\omega}_j^\top \boldsymbol{q}_t\right) \exp\left(-i\boldsymbol{\omega}_j^\top \boldsymbol{k}_\tau\right).$$

Finally, defining $\hat{\boldsymbol{Z}}_t^D = [\hat{z}_{t,1} \quad \hat{z}_{t,2} \quad \dots \quad \hat{z}_{t,D,}]$ we arrive at a similar result. The full recurrence of softmax attention, therefore, can be written as:

$$\hat{\boldsymbol{S}}_t^D = \hat{\boldsymbol{S}}_{t-1}^D \bar{\boldsymbol{R}}_t + \boldsymbol{v}_t \left(\phi(\boldsymbol{q}_t) \circ \phi(\boldsymbol{k}_t)\right)^\top, \quad \hat{\boldsymbol{Z}}_t^D = \hat{\boldsymbol{Z}}_{t-1}^D \bar{\boldsymbol{R}}_t + \phi(\boldsymbol{q}_t) \circ \phi(\boldsymbol{k}_t), \quad \hat{\boldsymbol{o}}_t = \frac{\hat{\boldsymbol{S}}_t^D \mathbf{1}}{\hat{\boldsymbol{z}}_t^D \mathbf{1}}.$$

which again highlights the importance of the gate $\bar{\boldsymbol{R}}$ as selective rotation.

### A.3   OPTIMAL VARIANCE FOR RANDOM FOURIER FEATURES

**Theorem 1** *Let the expected error of the RFF kernel over $\boldsymbol{\omega}_j \sim \mathcal{N}\left(0, \sigma^2 \boldsymbol{I}\right)$ be as follows:*
*$ERR\left[\boldsymbol{q}_t, \boldsymbol{k}_\tau\right] = \mathbb{E}_{\boldsymbol{\omega}_j}\left[\left(\frac{1}{D}\sum_{j=1}^{D} \phi_{\boldsymbol{\omega}_j}(\boldsymbol{q}_t) \cdot \phi_{\boldsymbol{\omega}_j}(\boldsymbol{k}_\tau) - \exp\left(\boldsymbol{q}_t^\top \boldsymbol{k}_\tau\right)\right)^2\right].$ Then, for a given a pair of*
*$L_2$ normalized query and key, the optimal value of $\sigma$ is equal to $\sigma = \tan\left(\frac{\arccos\left(\boldsymbol{q}_t^\top \boldsymbol{k}_\tau\right)}{2}\right).$*

**Proof 1** *We start by writing down the error:*

$$\begin{aligned}
ERR\left[\boldsymbol{q}_t, \boldsymbol{k}_\tau\right] =& \frac{e^2}{D^2} \sum_{j,j'=1} \mathbb{E}\left[\Re\left[\exp\left(i\left(\omega_j + \omega_{j'}\right)^\top \left(\boldsymbol{q}_t - \boldsymbol{k}_\tau\right)\right)\right]\right] \\
& - \frac{2e}{D} \sum_{j=1} \mathbb{E}\left[\Re\left[\exp\left(i\omega_j^\top \left(\boldsymbol{q}_t - \boldsymbol{k}_\tau\right)\right)\right]\right] \exp\left(\boldsymbol{q}_t^\top \boldsymbol{k}_\tau\right) + const. \\
=& \frac{e^2}{D} \mathbb{E}\left[\cos^2\left(i\omega^\top \left(\boldsymbol{q}_t - \boldsymbol{k}_\tau\right)\right)\right] + \frac{e^2\left(D^2 - D\right)}{D^2} \mathbb{E}\left[\cos\left(i\omega^\top \left(\boldsymbol{q}_t - \boldsymbol{k}_\tau\right)\right)\right]^2 \\
& - 2e \cdot \mathbb{E}\left[\cos\left(i\omega^\top \left(\boldsymbol{q}_t - \boldsymbol{k}_\tau\right)\right)\right] \exp\left(\boldsymbol{q}_t^\top \boldsymbol{k}_\tau\right) + const.,
\end{aligned}$$

*where the const. term corresponds to the terms constant w.r.t. the variance of the distribution $\sigma^2$. Plugging in the expectation of the $\cos(\cdot)$ and $\cos^2(\cdot)$ functions (Choromanski et al., 2021a), we get the following optimization problem:*

$$\min_{\sigma} \left[\frac{e^{2-4\sigma^2} \cdot \exp\left(-4\sigma^2\xi\right)}{2D} + \frac{D-1}{D}e^{2-2\sigma^2} \exp\left(-2\sigma^2\xi\right) - 2e^{1-\sigma^2} \exp\left(\left(1-\sigma^2\right)\xi\right)\right],$$

*where for simplicity, we set $\boldsymbol{q}_t^\top \boldsymbol{k}_\tau = \xi \in [0, 1]$. Since in most cases, $D$ is a sizable number, we try to solve this optimization problem in the limit $D \to \infty$, which is equivalent to:*

$$\min_{\sigma} \left[e^{2-2\sigma^2(1+\xi)} - 2e^{\left(1-\sigma^2\right)(1+\xi)}\right],$$

*with the optimal value equal to:*

$$\sigma = \sqrt{\frac{1-\xi}{1+\xi}}.$$

*Considering normalized queries and keys $||\boldsymbol{k}_t|| = ||\boldsymbol{q}_t|| = 1$ we can replace the $\xi = \boldsymbol{q}_t^\top \boldsymbol{k}_\tau$ with $\cos(\theta)$ therefore above also simplifies to:*

$$\sigma = \sqrt{\frac{1 - \cos(\theta)}{1 + \cos(\theta)}} = \tan(\theta/2).$$

*This completes our proof.* ∎

### A.3.1 PARAMETERIZATION OF THE TEMPERATURES

We can generalize the parameterization of our proposed temperatures vs. that of RoPE introduced by Su et al. (2021) as follows. Let $\epsilon$ be a small enough number. Then, we have:

$$\textbf{RoPE:} \quad \phi = \texttt{arange(0, D//2)} \qquad\qquad \Theta = \epsilon^\phi$$

$$\textbf{\textit{Selective RoPE:}} \quad \phi = \texttt{arange(0, D//2)} \cdot \frac{(1-\epsilon)\pi}{(D//2 - 1)} \qquad \Theta = \tan(\phi/2)$$

Here, $\epsilon$ can be seen as the inverse of the base frequency in RoPE (Su et al., 2021), and the upper-bound on the angle between the queries and keys in our temperature scheme. A visualization of the temperature distribution in *Selective RoPE* compared to standard *RoPE* is shown in Figure 2. Our proposed variation of the temperature has an extremely similar distribution, but with a slightly faster decay to 0.

### A.4 ROLE OF REAL AND IMAGINARY PARTS IN DIAGONAL SSMS

We start our analysis with non-selective diagonal SSMs and show the distinct roles of the real and imaginary components. SSMs can be derived from continuous-time representations, expressed as[1]:

$$\frac{d\boldsymbol{s}(t)}{dt} = \boldsymbol{A}\boldsymbol{s}(t) + \boldsymbol{k}v(t), \quad o(t) = \boldsymbol{q}^\top \boldsymbol{s}(t), \quad K(t) = \boldsymbol{q}^\top e^{\boldsymbol{A}t}\boldsymbol{k}, \quad o(t) = K(t) * v(t), \quad (20)$$

where we assume the continuous value signal $v(t)$ and the continuous output signal $o(t)$ to both be scalars. Inspired by S4D (Gu et al., 2022b), which is an SSM with diagonal $\boldsymbol{A}$, we initialize the imaginary part of the state matrix as $\boldsymbol{A}_n = i\omega_n$ ($n \in [0, N]$, roots of unity), from which the output is derived as:

$$o(t) = \sum_{n=1}^{N} \boldsymbol{k}_n \boldsymbol{q}_n e^{i\omega_n t} \int_{-\infty}^{\infty} e^{-i\omega_n \tau} v(\tau) u_t(\tau) d\tau, \quad u_t(\tau) = \begin{cases} 1, & 0 \leq \tau \leq t \\ 0, & \text{o.w.} \end{cases} \quad (21)$$

where $u_t(\tau)$ is a step-window function. The integral in Equation (21) is equivalent to computing the Fourier Transform of the windowed signal $v(\tau) u_t(\tau)$ at frequency $\omega_n$. Duality between convolution in the time domain and multiplication in the frequency domain simplifies eq. (21) to:

$$o(t) = \sum_{n=1}^{N} \boldsymbol{k}_n \boldsymbol{q}_n (V_{\omega_n} * U_{t,\omega_n}), \quad U_{t,2\omega} = \frac{\sin(\omega t)}{\omega} e^{-i\omega t} \quad (22)$$

with $V_{\omega_n}$ and $U_{t,\omega_n}$ denoting the Fourier transforms of $v(\tau)$ and $u_t(\tau)$, respectively. The input spectrum $V_\omega$ is convolved with the window spectrum $U_{t,\omega}$, causing distortion, a phenomenon known as *spectral leakage*. In the discrete domain, the integral in eq. (21) becomes a summation:

$$o_t = \sum_{n=0}^{N} \boldsymbol{q}_n \boldsymbol{k}_n \sum_{\tau=0}^{t} \exp\left(-\frac{2\pi i n\tau}{N}\right) v_\tau. \quad (23)$$

where $\omega_n = \frac{2\pi n i}{N}$ and $\Delta = \frac{1}{N}$. Thus, S4D with a purely imaginary state matrix $\boldsymbol{A}$ acts as a spectral analyzer: it accurately computes the $N$-point DFT of the value $v_t$ for $t \leq N$. But for $t > N$,

---

[1]For consistency within our notation, we replace the common SSM notation for the $\boldsymbol{B}$ and $\boldsymbol{C}$ matrix and the input with our self-attention based notation, i.e., $\boldsymbol{B}$ denoted as the key $\boldsymbol{k}$, $\boldsymbol{C}$ denoted as the query $\boldsymbol{q}$, and the input signal $u$ denoted as the value $v$. For a detailed comparison, refer to Table 2 from Yang et al. (2024b).

this spectral analysis suffers from **spectral leakage** since the state size can at most represent $N$ frequencies. Therefore, the higher frequencies are being aliased or overwritten.

In Signal Processing, spectral leakage is addressed by windowing (Harris, 2005). In S4D, this is achieved implicitly by using a complex state matrix $\boldsymbol{A}$ with the real part acting as a ***window function***, a classical solution to spectral leakage (Oppenheim, 1999). Concretely, with $\boldsymbol{A} = \exp(-\alpha_n\Delta + 2\pi in\Delta)$, S4D performs a windowed DFT using a *Poisson window* (III, 2011), thereby avoiding spectral leakage. Its output can be written as:

$$o_t = \sum_{n=0}^{N} \boldsymbol{q}_n \boldsymbol{k}_n \sum_{\tau=0}^{t} \exp\left(-\tfrac{2\pi in\tau}{N}\right) v_\tau \underbrace{\exp(-\alpha_n\Delta\tau)}_{w_\tau}, \tag{24}$$

where $w_\tau$ is the Poisson window and $\Delta = \frac{1}{N}$ is chosen for clarity in the DFT formulation. Thus, the real part of $\boldsymbol{A}$ in S4D acts as a window, suppressing spectral leakage and enabling undistorted spectral representations. Therefore, to summarize: *the two real and imaginary parts of state transition matrix $\boldsymbol{A}$ serve distinct but complementary roles; **Imaginary** parts extract spectral information, while **Real** parts suppress leakage and ensure clean representation of the spectrum.*

### A.5 COMPLEX ROTATIONS AND HOUSEHOLDER MATRICES

Another approach towards introducing rotations to the queries and keys is using Householder reflection matrices (Yang et al., 2024b; 2025b). In this approach, the rotation of the query and key pair is limited to a single reflection along the direction of an input-dependent vector. Specifically, let $\boldsymbol{w}_t$ be an input-dependent unit vector. Then, the positional information is encoded through the product of Householder reflection matrices as:

$$\boldsymbol{q}_t^\top \boldsymbol{R}_{t:\tau} \boldsymbol{k}_\tau = \boldsymbol{q}_t^\top \left( \prod_{\kappa=\tau+1}^{t} \left( \boldsymbol{I} - 2\beta_\kappa \cdot \boldsymbol{w}_\kappa \boldsymbol{w}_\kappa^\top \right) \right) \boldsymbol{k}_\tau.$$

Therefore, the positional information between the $t^{th}$ and $\tau^{th}$ token is encoded through a rotation consisting of $t - \tau$ reflections.

Conveniently, we can also write the complex diagonal rotation matrix in *Selective RoPE* in terms of the product of Householder matrices. Specifically, we can write the realification of the rotation matrix $\boldsymbol{R_t}$ as the product of $d$ Householder reflections, each of which performs the reflection over a single pair of adjacent elements:

$$\boldsymbol{R}_t = \prod_{j=1}^{d} \left( \boldsymbol{I} - 2 \cdot \begin{bmatrix} \boldsymbol{0}_j \\ 1 \\ 0 \\ \boldsymbol{0}_{d-j-2} \end{bmatrix} \begin{bmatrix} \boldsymbol{0}_j \\ 1 \\ 0 \\ \boldsymbol{0}_{d-j-2} \end{bmatrix}^\top \right) \left( \boldsymbol{I} - 2 \begin{bmatrix} \boldsymbol{0}_j \\ \cos(\omega_{t,j}/2) \\ \sin(\omega_{t,j}/2) \\ \boldsymbol{0}_{d-j-2} \end{bmatrix} \begin{bmatrix} \boldsymbol{0}_j \\ \cos(\omega_{t,j}/2) \\ \sin(\omega_{t,j}/2) \\ \boldsymbol{0}_{d-j-2} \end{bmatrix}^\top \right),$$

where we define $\boldsymbol{0}_m \in \mathbb{R}^m$ as a vector with all zeros. Assuming we split adjacent elements in the query-key into the real and imaginary components, then *Selective RoPE* is performing two reflections over each adjacent element pair of the input, with one of them a parametric reflection, and the other negating the first element.

This interpretation also explains why we gain more expressivity when using *Selective RoPE*: due to the block-diagonal structure, there is a channel mixing happening between the adjacent query-key elements. Channel mixing is a key component in improving the expressivity of sequence models (Cirone et al., 2024), thus improving the state-tracking abilities of the network (Siems et al., 2025).

### A.6 KERNELIZED LINEAR ATTENTION WITH FIXED PARAMETERS

In order to further contextualize our results for the RFF kernel and its relationship to other variants of kernelized linear attention, we try to replicate our mathematical derivations for two more variants of these models: namely Performer and CosFormer (Choromanski et al., 2021b; Qin et al., 2022). We then present the results side-by-side to provide further insights into the importance of complex parameterization in Linear attention. We provide a summary of the models in Table 4.

| Model | Transition Matrix |
|---|---|
| *Input-dependent* | |
| Performer (PRF-based) | $\bar{\boldsymbol{R}}_t = \mathrm{diag}\big(\exp(\boldsymbol{\Omega}^\top(\boldsymbol{q}_t - \boldsymbol{q}_{t-1}))\big)$ |
| RFA (RFF-based) | $\bar{\boldsymbol{R}}_t = \mathrm{diag}\big(\exp(i\,\boldsymbol{\Omega}^\top(\boldsymbol{q}_t - \boldsymbol{q}_{t-1}))\big)$ |
| *Input-independent* | |
| RoFormer (RoPE) | $\bar{\boldsymbol{R}} = \mathrm{diag}(\exp(i\,\boldsymbol{\omega}))$ |
| CosFormer (diagonal RoPE) | $\bar{\boldsymbol{R}} = \exp(i\,\omega)\boldsymbol{I}$ |

Table 4: Kernelized linear attention models under the shared recurrence $\boldsymbol{S}_t = \boldsymbol{S}_{t-1}\boldsymbol{A}_t + \boldsymbol{v}_t\boldsymbol{k}_t^\top$. The top block lists input-dependent transitions, while the bottom block lists input-independent ones. Among these, the Performer transition is not norm-preserving in general.

### A.6.1 PERFORMER

We use the notation provided in Appendix A.2. To approximate the exponential kernel $\exp(\cdot)$, we use Positive Random Features (PRF) (Rahimi & Recht, 2007; Choromanski et al., 2021b) with random features $\boldsymbol{\omega} \in \mathbb{R}^d \sim \mathcal{N}(0, \sigma^2\boldsymbol{I})$. The feature map is defined as

$$\phi_{\boldsymbol{\omega}}(\boldsymbol{x}) = \exp\left(-\frac{\|\boldsymbol{x}\|_2^2}{2} + \boldsymbol{\omega}^\top\boldsymbol{x}\right),$$

so that

$$\exp(\boldsymbol{q}_t^\top\boldsymbol{k}_\tau) = \mathbb{E}_{\boldsymbol{\omega}\sim\mathcal{N}(0,\sigma^2\boldsymbol{I})}\big[\phi_{\boldsymbol{\omega}}(\boldsymbol{q}_t)^\top\phi_{\boldsymbol{\omega}}(\boldsymbol{k}_\tau)\big],$$

for $\sigma = 1$. By applying this feature map, the linear attention formulation in Equation (2), we can approximate the exponential kernel in softmax attention. Continuing the approximation:

$$\exp(\boldsymbol{q}_t^\top\boldsymbol{k}_\tau) = \exp\left(-\frac{\|\boldsymbol{q}_t\|_2^2 + \|\boldsymbol{k}_\tau\|_2^2}{2}\right) \cdot \mathbb{E}_{\boldsymbol{\omega}\sim\mathcal{N}(0,\boldsymbol{I})}\big[\exp(\boldsymbol{\omega}^\top\boldsymbol{q}_t)\exp(\boldsymbol{\omega}^\top\boldsymbol{k}_\tau)\big].$$

Let $\boldsymbol{\omega}_j \sim \mathcal{N}\big(0, \sigma^2\boldsymbol{I}\big)$ for $j \in \{1, 2, \ldots, D\}$. Then due to the law of large numbers we have:

$$\exp(\boldsymbol{q}_t^\top\boldsymbol{k}_\tau) = \exp\left(-\frac{\|\boldsymbol{q}_t\|_2^2 + \|\boldsymbol{k}_\tau\|_2^2}{2}\right) \cdot \lim_{D\to\infty}\frac{1}{D}\sum_{j=1}^{D}\exp(\boldsymbol{\omega}_j^\top\boldsymbol{q}_t)\cdot\exp(\boldsymbol{\omega}_j^\top\boldsymbol{k}_\tau).$$

Therefore, we can approximate $\exp(\boldsymbol{q}_t^\top\boldsymbol{k}_\tau)$ as the dot product of the random exponential projection of the query and the key using $D$ random $\boldsymbol{\omega}_j$s:

$$\hat{\boldsymbol{s}}_t^D = \frac{1}{D}\sum_{\tau=1}^{t}\sum_{j=1}^{D}\exp\left(-\frac{\|\boldsymbol{q}_t\|_2^2 + \|\boldsymbol{k}_\tau\|_2^2}{2}\right)\exp(\boldsymbol{\omega}_j^\top\boldsymbol{q}_t)\exp(\boldsymbol{\omega}_j^\top\boldsymbol{k}_\tau)\cdot\boldsymbol{v}_\tau.$$

This allows us to compute the softmax attention as the linear attention parameterized by:

$$\phi(\boldsymbol{q}_t) = \exp\left(-\frac{\|\boldsymbol{q}_t\|_2^2}{2}\right) \cdot \exp(\boldsymbol{\Omega}^\top\boldsymbol{q}_t), \quad \phi(\boldsymbol{k}_\tau) = \exp\left(-\frac{\|\boldsymbol{k}_t\|_2^2}{2}\right) \cdot \exp(\boldsymbol{\Omega}^\top\boldsymbol{k}_\tau),$$

with $\lim_{D\to\infty}\Re\big\{\hat{\boldsymbol{s}}_t^D\big\} = \sum_{\tau=1}^{t}\exp(\boldsymbol{q}_t^\top\boldsymbol{k}_\tau)\cdot\boldsymbol{v}_\tau$ and $\boldsymbol{\Omega} = [\boldsymbol{\omega}_1, ..., \boldsymbol{\omega}_D]$. Omitting the superscript $D$ for simplifying the notation, let us focus on one random feature $\boldsymbol{\omega}_j$ and its contribution to the output:

$$\hat{\boldsymbol{s}}_{t,j} = \sum_{\tau=1}^{t}\exp\left(-\frac{\|\boldsymbol{q}_t\|_2^2}{2}\right)\exp\left(-\frac{\|\boldsymbol{k}_\tau\|_2^2}{2}\right)\exp(\boldsymbol{\omega}_j^\top\boldsymbol{q}_t)\exp(\boldsymbol{\omega}_j^\top\boldsymbol{k}_\tau)\cdot\boldsymbol{v}_\tau.$$

In this case, we have $\hat{\boldsymbol{s}}_t^D = \frac{1}{D}\hat{\boldsymbol{S}}_t^D\boldsymbol{1}$, where $\hat{\boldsymbol{S}}_t^D = [\hat{\boldsymbol{s}}_{t,1} \quad \hat{\boldsymbol{s}}_{t,2} \quad \ldots \quad \hat{\boldsymbol{s}}_{t,D}] \in \mathbb{R}^{d\times D}$. Now note that we have:

$$\hat{\boldsymbol{s}}_{t,j} = \sum_{\tau=1}^{t-1}\exp\left(-\frac{\|\boldsymbol{q}_t\|_2^2 - \|\boldsymbol{q}_{t-1}\|_2^2}{2}\right)\exp(\boldsymbol{\omega}_j^\top\boldsymbol{q}_{t-1})\exp(\boldsymbol{\omega}_j^\top(\boldsymbol{q}_t - \boldsymbol{q}_{t-1}))\exp(\boldsymbol{\omega}_j^\top\boldsymbol{k}_\tau)\cdot\boldsymbol{v}_\tau \quad (25)$$

$$+ \exp\left(-\frac{\|\boldsymbol{q}_t\|_2^2}{2}\right)\exp\left(-\frac{\|\boldsymbol{k}_t\|_2^2}{2}\right)\exp(\boldsymbol{\omega}_j^\top(\boldsymbol{q}_t - \boldsymbol{k}_t))\cdot\boldsymbol{v}_t. \quad (26)$$

$$= \exp\left(-\frac{\|\boldsymbol{q}_t\|_2^2 - \|\boldsymbol{q}_{t-1}\|_2^2}{2}\right)\exp(\boldsymbol{\omega}_j^\top(\boldsymbol{q}_t - \boldsymbol{q}_{t-1}))\,\hat{\boldsymbol{s}}_{t-1}^j + \phi_{\boldsymbol{\omega}_j}(\boldsymbol{q}_t)\cdot\phi_{\boldsymbol{\omega}_j}(\boldsymbol{k}_t)\cdot\boldsymbol{v}_t \quad (27)$$

Note that the real exponential component in Equation ($27$) can introduce instability to the recurrence. Therefore, following the standard in both linear Transformers (Yang et al., 2024b;a; 2025a; Lin et al., 2025) and deep softmax Transformers (Henry et al., 2020), we assume $L_2$ normalization over the query and the key, i.e., $\|\boldsymbol{q}_t\|_2 = \|\boldsymbol{q}_{t-1}\|_2$. Thus, recurrence presented in Equation ($27$) simplifies to:

$$\hat{\boldsymbol{s}}_{t,j} = \exp\big(\boldsymbol{\omega}_j^\top (\boldsymbol{q}_t - \boldsymbol{q}_{t-1})\big) \hat{\boldsymbol{s}}_{t-1,j} + \phi_{\boldsymbol{\omega}_j}(\boldsymbol{q}_t) \cdot \phi_{\boldsymbol{\omega}_j}(\boldsymbol{k}_t) \cdot \boldsymbol{v}_t,$$

with $\hat{\boldsymbol{s}}_{t,j}$ being the $j^{th}$ column of $\hat{\boldsymbol{S}}_t^D$ is scaled by the values $\exp\big(\boldsymbol{\omega}_j^\top (\boldsymbol{q}_t - \boldsymbol{q}_{t-1})\big)$. Therefore, we can write the recurrence over $\hat{\boldsymbol{S}}_t$ as:

$$\hat{\boldsymbol{S}}_t^D = \hat{\boldsymbol{S}}_{t-1} \bar{\boldsymbol{R}}_t + \boldsymbol{v}_t \left(\phi(\boldsymbol{q}_t) \circ \phi(\boldsymbol{k}_t)\right)^\top, \quad \hat{\boldsymbol{s}}_t^D = \frac{1}{D} \hat{\boldsymbol{S}}_t^D \mathbf{1}.$$

where $\phi(x)$ is a vector with its $j^{th}$ element equal to $\phi_{\boldsymbol{\omega}_j}(x)$, and $\bar{\boldsymbol{R}}_t$ is:

$$\bar{\boldsymbol{R}}_t = \mathrm{diag}(\exp(\boldsymbol{\Omega}^\top (\boldsymbol{q}_t - \boldsymbol{q}_{t-1}))) \tag{28}$$

Focusing on Equation ($28$), we observe that the PRF approximation no-longer results in an input-depedent rotation matrix, but rather a more general form of a real-valued input-dependent state transition matrix. However, the features $\boldsymbol{\Omega}$ are randomly sampled from a normal distribution.

Similarly, we can also approximate the normalizing factor $\boldsymbol{z}_t$ as:

$$\hat{\boldsymbol{z}}_t^D = \frac{1}{D} \sum_{\tau=1}^{t} \sum_{j=1}^{D} \exp\left(-\frac{\|\boldsymbol{q}_t\|_2^2 + \|\boldsymbol{k}_\tau\|_2^2}{2}\right) \exp\big(\boldsymbol{\omega}_j^\top \boldsymbol{q}_t\big) \exp\big(\boldsymbol{\omega}_j^\top \boldsymbol{k}_\tau\big).$$

Separating the contribution of each random feature, we have:

$$\hat{\boldsymbol{z}}_{t,j} = \sum_{\tau=1}^{t} \exp\left(-\frac{\|\boldsymbol{q}_t\|_2^2}{2}\right) \exp\left(-\frac{\|\boldsymbol{k}_\tau\|_2^2}{2}\right) \exp\big(\boldsymbol{\omega}_j^\top \boldsymbol{q}_t\big) \exp\big(\boldsymbol{\omega}_j^\top \boldsymbol{k}_\tau\big).$$

Finally, defining $\hat{\boldsymbol{Z}}_t^D = [\hat{\boldsymbol{z}}_{t,1} \quad \hat{\boldsymbol{z}}_{t,2} \quad \ldots \quad \hat{\boldsymbol{z}}_{t,D,}]$ we arrive at a similar result. The full recurrence of softmax attention, therefore, can be written as:

$$\hat{\boldsymbol{S}}_t^D = \hat{\boldsymbol{S}}_{t-1}^D \bar{\boldsymbol{R}}_t + \boldsymbol{v}_t \left(\phi(\boldsymbol{q}_t) \circ \phi(\boldsymbol{k}_t)\right)^\top, \quad \hat{\boldsymbol{Z}}_t^D = \hat{\boldsymbol{Z}}_{t-1}^D \bar{\boldsymbol{R}}_t + \phi(\boldsymbol{q}_t) \circ \phi(\boldsymbol{k}_t), \quad \hat{\boldsymbol{o}}_t = \frac{\hat{\boldsymbol{S}}_t^D \mathbf{1}}{\hat{\boldsymbol{z}}_t^D \mathbf{1}}.$$

Interestingly, unlike the results provided in Appendix A.3, in this case we won't be able to rely on a spectrum of choices for $\sigma$ in order to limit the approximation error for a finite number of random feature samples. More concretely, we provide the following theorem to highlight this issue:

**Theorem 2** *Let the expected error of the PRF kernel over $\boldsymbol{\omega}_j \sim \mathcal{N}\left(0, \sigma^2 \boldsymbol{I}\right)$ be as follows: $ERR\left[\boldsymbol{q}_t, \boldsymbol{k}_\tau\right] = \mathbb{E}_{\boldsymbol{\omega}_j}\left[\left(\frac{1}{D}\sum_{j=1}^{D} \phi_{\boldsymbol{\omega}_j}(\boldsymbol{q}_t) \cdot \phi_{\boldsymbol{\omega}_j}(\boldsymbol{k}_\tau) - \exp\big(\boldsymbol{q}_t^\top \boldsymbol{k}_\tau\big)\right)^2\right]$. Then, for a given a pair of $L_2$ normalized query and key, the optimal value of $\sigma$ is equal to $\sigma = 1$.*

**Proof 2** *We start by writing down the error:*

$$ERR\left[\boldsymbol{q}_t, \boldsymbol{k}_\tau\right] = \frac{e^{-2}}{D^2} \sum_{j,j'=1} \mathbb{E}\left[\exp\left(\left(\omega_j + \omega_{j'}\right)^\top (\boldsymbol{q}_t - \boldsymbol{k}_\tau)\right)\right]$$
$$- \frac{2e^{-1}}{D} \sum_{j=1} \mathbb{E}\left[\exp\big(\omega_j^\top (\boldsymbol{q}_t - \boldsymbol{k}_\tau)\big)\right] \exp\big(\boldsymbol{q}_t^\top \boldsymbol{k}_\tau\big) + const.$$
$$= \frac{e^{-2}}{D} \mathbb{E}\left[\exp\big(2\omega^\top (\boldsymbol{q}_t + \boldsymbol{k}_\tau)\big)\right] + \frac{e^{-2}\left(D^2 - D\right)}{D^2} \mathbb{E}\left[\exp\big(\omega^\top (\boldsymbol{q}_t + \boldsymbol{k}_\tau)\big)\right]^2$$
$$- 2e^{-1} \cdot \mathbb{E}\left[\exp\big(\omega^\top (\boldsymbol{q}_t + \boldsymbol{k}_\tau)\big)\right] \exp\big(\boldsymbol{q}_t^\top \boldsymbol{k}_\tau\big) + const.,$$

*where the const. term corresponds to the terms constant w.r.t. the variance of the distribution $\sigma^2$. Plugging in the expectation of the $\exp(\cdot)$ function (Choromanski et al., 2021a), we get the following optimization problem:*

$$\min_\sigma \left[\frac{e^{-2}}{D} \exp\big(4\sigma^2 (1 + \xi)\big) + \frac{e^{-2}(D-1)}{D} \exp\big(2\sigma^2 (1 + \xi)\big) - 2e^{-1} \exp\big(\sigma^2 (1 + \xi) + \xi\big)\right],$$

*where for simplicity, we set $\boldsymbol{q}_t^\top \boldsymbol{k}_\tau = \xi \in [0, 1]$. Since in most cases, $D$ is a sizable number, we try to solve this optimization problem in the limit $D \to \infty$, which is equivalent to:*

$$\min_\sigma \left[ e^{2\sigma^2(1+\xi)-2} - 2e^{\sigma^2(1+\xi)+\xi-1} \right],$$

*with the optimal value equal to $\sigma = 1$.*

*This completes our proof.* ∎

Following Theorem 2, we observe that for a fixed pair of queries and keys, the optimal variance is independent of the angle between the two vectors. This is in stark contrast of the RFF-related results in Theorem 1, where the variance of the random feature is a function of the angle between the query and key vectors.

At first glance, this may seem as a good incentive to instead adopt the transition matrix introduced in Equation (28) for the recurrence. However, one also need to consider that there is no stable generalization of a recurrence based on this transition matrix, as there won't be any guarantees about the matrix being contractive or orthogonal. On the other hand, the rotation matrix in Equation (19) will always be stable, as it corresponds to an orthogonal matrix. This makes RFFs a much more suitable choice for a random features-based approximation of the softmax attention function.

### A.6.2 COSFORMER

We start by defining the output in the CosFormer model (Qin et al., 2022):

$$\boldsymbol{o}_t = \frac{\boldsymbol{S}_t \boldsymbol{q}_t}{\boldsymbol{z}_t^\top \boldsymbol{q}_t}, \quad \boldsymbol{S}_t = \sum_{\tau=1}^{t} \cos\left(\frac{\pi(t-\tau)}{2M}\right) \cdot \boldsymbol{v}_\tau \boldsymbol{k}_\tau^\top, \quad \boldsymbol{z}_t = \sum_{\tau=1}^{t} \cos\left(\frac{\pi(t-\tau)}{2M}\right) \cdot \boldsymbol{k}_\tau, \quad (29)$$

where $M$ corresponds to a pre-defined, large scalar value. First, we focus on the state $\boldsymbol{S}_t$ in Equation (29), which we can write as:

$$\boldsymbol{S}_t = \cos\left(\frac{\pi}{2M}\right) \cdot \sum_{\tau=1}^{t-1} \cos\left(\frac{(t-1-\tau)\pi}{2M}\right) \cdot \boldsymbol{v}_\tau \boldsymbol{k}_\tau^\top$$

$$- \sin\left(\frac{(t-1-\tau)\pi}{2M}\right) \cdot \sum_{\tau=1}^{t-1} \sin\left(\frac{(t-1-\tau)\pi}{2M}\right) \cdot \boldsymbol{v}_\tau \boldsymbol{k}_\tau^\top + \boldsymbol{v}_t \boldsymbol{k}_t^\top$$

$$= \cos\left(\frac{\pi}{2M}\right) \cdot \boldsymbol{S}_{t-1} - \sin\left(\frac{\pi}{2M}\right) \cdot \boldsymbol{S}_{t-1}^{\sin} + \boldsymbol{v}_t \boldsymbol{k}_t^\top,$$

where we define:

$$\boldsymbol{S}_t^{\sin} = \sum_{\tau=1}^{t} \sin\left(\frac{(t-\tau)\pi}{2M}\right) \cdot \boldsymbol{v}_\tau \boldsymbol{k}_\tau^\top$$

$$= \sin\left(\frac{\pi}{2M}\right) \cdot \sum_{\tau=1}^{t-1} \cos\left(\frac{(t-1-\tau)\pi}{2M}\right) \cdot \boldsymbol{v}_\tau \boldsymbol{k}_\tau^\top + \cos\left(\frac{\pi}{2M}\right) \cdot \sum_{\tau=1}^{t-1} \sin\left(\frac{(t-1-\tau)\pi}{2M}\right) \cdot \boldsymbol{v}_\tau \boldsymbol{k}_\tau^\top$$

$$= \sin\left(\frac{\pi}{2M}\right) \boldsymbol{S}_{t-1} + \cos\left(\frac{\pi}{2M}\right) \boldsymbol{S}_{t-1}^{\sin}.$$

Now, let us define:

$$\tilde{\boldsymbol{S}}_t = \begin{bmatrix} \boldsymbol{S}_t & \boldsymbol{S}_t^{\sin} \end{bmatrix},$$

for which we can write the following recurrence:

$$\tilde{\boldsymbol{S}}_t = \tilde{\boldsymbol{S}}_{t-1} \begin{bmatrix} \cos\left(\frac{\pi}{2M}\right)\boldsymbol{I} & \sin\left(\frac{\pi}{2M}\right)\boldsymbol{I} \\ -\sin\left(\frac{\pi}{2M}\right)\boldsymbol{I} & \cos\left(\frac{\pi}{2M}\right)\boldsymbol{I} \end{bmatrix} + \begin{bmatrix} \boldsymbol{v}_t \boldsymbol{k}_t^\top & \boldsymbol{0} \end{bmatrix}.$$

This recurrence is equivalent to a complex-domain recurrence where $\boldsymbol{S}_t$ corresponds to the real component of the state and $\boldsymbol{S}_t^{\sin}$ to the imaginary component:

$$\tilde{\boldsymbol{S}}_t = \tilde{\boldsymbol{S}}_{t-1}\bar{\boldsymbol{R}} + \boldsymbol{v}_t \boldsymbol{k}_t^\top, \quad \bar{\boldsymbol{R}} = \exp\left(i\frac{\pi}{2M}\right)\boldsymbol{I}. \tag{30}$$

Similarly, we can define the complex-domain normalization factor $\tilde{\boldsymbol{z}}_t$ as:

$$\tilde{\boldsymbol{z}}_t = \begin{bmatrix} \boldsymbol{z}_t & \boldsymbol{z}_t^{\sin} \end{bmatrix},$$

where we define $\boldsymbol{z}_t^{\sin}$ as:

$$\boldsymbol{z}_t^{\sin} = \sin\left(\frac{\pi}{2M}\right)\boldsymbol{z}_{t-1} + \cos\left(\frac{\pi}{2M}\right)\boldsymbol{z}_{t-1}^{\sin}.$$

This recurrence is also equivalent to a complex-domain recurrence of the following form:

$$\tilde{\boldsymbol{z}}_t = \tilde{\boldsymbol{z}}_{t-1}\bar{\boldsymbol{R}} + \boldsymbol{k}_t, \quad \boldsymbol{R} = \exp\left(i\frac{\pi}{2M}\right)\boldsymbol{I}.$$

The model introduced in Equation (29) is a complex-domain linear attention model, in which the rotation matrix is defined as a diagonal matrix with the same values on the diagonal (see Equation (30)). Note that as demonstrated by Qin et al. (2022), the rope trick can be applied to further simplify the computation.

### A.7 RELATIONSHIP BETWEEN *Selective RoPE*, ALiBi, AND FoX

The use of exponential decay in softmax Transformer has been common practice for a while (Press et al., 2021; Jelassi et al., 2024; Lin et al., 2025), proving advantageous for length generalization and position encoding. In this section, we investigate ALiBi and FoX as two examples of such models and provide context for their success based on the findings of this paper.

#### A.7.1 ATTENTION WITH LINEAR BIASES (ALiBi)

ALiBi (Press et al., 2021) was originally introduced as a position encoding technique aiming at improving the length generalization of softmax Transformers. Borrowing the notation from Equation (1), we write the attention mechanism in ALiBi as:

$$\boldsymbol{s}_t = \sum_{\tau=1}^{t} \exp\left(\frac{1}{\sqrt{d}}\boldsymbol{q}_t^\top \boldsymbol{k}_\tau + \sum_{\kappa=\tau+1}^{t} \log \boldsymbol{A}\right)\boldsymbol{v}_\tau, \quad \boldsymbol{z}_t = \sum_{\tau=1}^{t} \exp\left(\frac{1}{\sqrt{d}}\boldsymbol{q}_t^\top \boldsymbol{k}_\tau + \sum_{\kappa=\tau+1}^{t} \log \boldsymbol{A}\right),$$

where $\log \boldsymbol{A} < 0$ can be a scalar or a diagonal matrix. Note that we can re-write $\boldsymbol{s}_t$ and $\boldsymbol{z}_t$ by moving the input-independent summation outside of the exponent:

$$\boldsymbol{s}_t = \sum_{\tau=1}^{t} \exp\left(\frac{1}{\sqrt{d}}\boldsymbol{q}_t^\top \boldsymbol{k}_\tau\right) \cdot \boldsymbol{A}^{t-\tau}\boldsymbol{v}_\tau, \qquad \boldsymbol{z}_t = \sum_{\tau=1}^{t} \exp\left(\frac{1}{\sqrt{d}}\boldsymbol{q}_t^\top \boldsymbol{k}_\tau\right) \cdot \boldsymbol{A}^{t-\tau},$$

which can be interpreted as a Transformer layer with exponential decay.

#### A.7.2 FORGETTING TRANSFORMERS (FoX)

FoX (Lin et al., 2025) is a softmax Transformer that aims to augment the attention mechanism with a forget gate, inspired by the success of the exponentially decaying gating mechanisms in linear RNNs and linear Transformers (Yang et al., 2024a; Gu & Dao, 2023). Borrowing the notation from Equation (1), we write the attention mechanism in FoX as:

$$\boldsymbol{s}_t = \sum_{\tau=1}^{t} \exp\left(\frac{1}{\sqrt{d}}\boldsymbol{q}_t^\top \boldsymbol{k}_\tau + \sum_{\kappa=\tau+1}^{t} \log \boldsymbol{A}_\kappa\right)\boldsymbol{v}_\tau, \quad \boldsymbol{z}_t = \sum_{\tau=1}^{t} \exp\left(\frac{1}{\sqrt{d}}\boldsymbol{q}_t^\top \boldsymbol{k}_\tau + \sum_{\kappa=\tau+1}^{t} \log \boldsymbol{A}_\kappa\right),$$

where $\log \boldsymbol{A}_\kappa < 0$ can be a scalar or a diagonal matrix. In this case, $\boldsymbol{A}_\kappa$ is an input dependent factor and a function of the $\kappa^{th}$ token. We can re-write $\boldsymbol{s}_t$ and $\boldsymbol{z}_t$ by moving the input-dependent summation outside of the exponent:

$$\boldsymbol{s}_t = \sum_{\tau=1}^{t} \exp\left(\tfrac{1}{\sqrt{d}}\boldsymbol{q}_t^\top \boldsymbol{k}_\tau\right) \cdot \prod_{\kappa=\tau+1}^{t} \boldsymbol{A}_\kappa \boldsymbol{v}_\tau, \qquad \boldsymbol{z}_t = \sum_{\tau=1}^{t} \exp\left(\tfrac{1}{\sqrt{d}}\boldsymbol{q}_t^\top \boldsymbol{k}_\tau\right) \cdot \prod_{\kappa=\tau+1}^{t} \boldsymbol{A}_\kappa,$$

which can be interpreted as a Transformer layer with selective exponential decay.

### A.7.3 THE COMPLEMENTARY ROLE OF ROTATION AND DECAY

The success of models belonging to the family of softmax Transformers with forget gates, namely ALiBi and FoX, demonstrate that the exponential decaying factor is a missing component in these models. This observation aligns with our proposed unifying framework in Section 3. Interestingly, in the softmax setting, *Selective RoPE* closely parallels FoX: it can be seen as endowing the decay term $a_t$ with a rotation component $\boldsymbol{R}_t$.

## B EXPERIMENTAL DETAILS

In this section we provide additional details on our experimental setup for the tasks considered in the paper.

### B.1 LANGUAGE MODELING

We use PlainLM (Ajroldi, 2024) together with an adapted version of `flash-linear-attention` for all of our language model trainings. We train on $> 80$GB VRAM GPUs including NVIDIA A100, H100 and B200. One model training (370M parameters, 10B tokens) is performed on a single node with 4 to 8 of such GPUs and takes anywhere from 48 hours (on 4 A100) to 9 hours on 8 B200. We use Distributed Data Parallel (DDP) for multi-GPU training.

Table 5: Optimizer and learning-rate schedule hyperparameters for language modeling.

| Parameter | Symbol | Value |
|---|---|---|
| **Optimizer** | | |
| Base learning rate (candidates) | $\eta$ | `[5e-4, 1e-3, 2e-3, 4e-3, 8e-3, 1.6e-2]` |
| Adam $\beta_1$ | $\beta_1$ | 0.9 |
| Adam $\beta_2$ | $\beta_2$ | 0.95 |
| Weight decay | $\lambda$ | 0.1 |
| Numerical epsilon | $\epsilon$ | $1 \times 10^{-8}$ |
| Gradient clipping (global norm) | $\mathrm{clip}_{\ell_2}$ | 1.0 |
| **LR Schedule / Training Horizon** | | |
| LR start (schedule) | $\eta_{\mathrm{start}}$ | 1e-5 |
| LR end (schedule) | $\eta_{\mathrm{end}}$ | 1e-4 |
| Warmup (fraction of steps) | – | 0.1 |
| Total optimizer steps | $T$ | 66,758 |

### B.2 SYNTHETIC TASKS

#### B.2.1 MAD

For MAD, we take the implementation from `mad_lab` and implement *Selective RoPE* in GLA. We follow the exact experimental setup outlined in the paper (Poli et al., 2024) and run all variations of task difficulty and optimizer hyperparameters which results in 66 task settings $\times$ 6 optimizer settings $= 396$ trained models per considered setting (i.e., GLA with *Selective RoPE*, RoPE or NoPE). We provide the logs from the experiments in our supplementary.

#### B.2.2 STATE TRACKING

For state tracking we adopt the exact experimental setup as described in DeltaProduct (Siems et al., 2025) and Grazzi et al. (2025).

Table 6: Training state tracking configuration.

| Training Loop | |
|---|---|
| Parameter | Value |
| Epochs | 100 |
| Batch size | 4096 |
| **Optimization** | |
| Learning rate | 1e-3 |
| $\beta_1$ | 0.9 |
| $\beta_2$ | 0.999 |
| Optimizer $\epsilon$ | 1e-8 |
| Weight decay | 1e-6 |
| LR scheduler | cosine |
| **Precision / Compile** | |
| Mixed precision | true |
| DType | bfloat16 |
| **Data** | |
| Train set size | 2,000,000 sequences |
| Train sequence length | 128 tokens |
| Eval set size | 500,000 sequences |
| Eval sequence length | 512 tokens |
| **Seeds & Eval** | |
| Seeds | [555, 666, 777, 888, 999] |
| Eval batch size | 128 |

### B.2.3 MQAR

We have carefully followed the training recipe of Arora et al. (2024a) for all models including: GLA (Yang et al., 2024a), DeltaNet (Yang et al., 2024b), Mamba2 (Dao & Gu, 2024) and Transformer++ (Touvron et al., 2023). The learning rate for all models was swept within the range of $[0.0001, 0.01]$ for 8 different values per each model ranging uniformly from 0.01 to 0.001. All other configuration and the model dimensions were remained the same as original reference Arora et al. (2024a).

### B.2.4 COPYING

### THE USE OF LARGE LANGUAGE MODELS (LLMS)

While preparing this manuscript, we used Large Language Models (LLMs) to a limited extent. Their role was restricted to assisting with editing and polishing the writing, such as improving clarity, grammar, and flow. All conceptual ideas, methods, experiments, and analyses presented in this paper are entirely the work of the authors. No ideas, algorithms, or research contributions were generated by an LLM. The LLM served only as a tool to refine the presentation of the text without influencing the substance of the research.

Table 7: Optimizer and Data parameters for Copying

| Optimizer | |
|---|---|
| Learning rate | 5.0e-5 |
| Weight decay | 0.1 |
| $\beta_1$ | 0.9 |
| $\beta_2$ | 0.999 |
| Optimizer $\epsilon$ | 1.0e-8 |
| Gradient clipping (global norm) | 1.0 |
| **Scheduler** | |
| Scheduler | linear |
| Warmup (fraction of steps) | 0.1 |
| **Seeds & Eval** | |
| Seed | 42 |
| Eval batch size | 256 |
| **Data** | |
| Vocab size | 26 |
| $n$-gram | 0 |
| Answer length | 0 |
| Train task | `copy` |
| Eval task | `copy` |
| Sequence length | 420 |
| Min length (train) | 2 |
| Max length (train) | 64 |
| Min length (eval) | 2 |
| Max length (eval) | 512 |
| Sampler type | sequential |
| Sampler seed | null |

