# OpenReview forum: "Selective Rotary Position Embedding"
_ICLR.cc/2026/Conference — ICLR 2026 Poster_

### Official Review · Reviewer_Rgdu · 2025-10-29

**Soundness:** 2
**Presentation:** 2
**Contribution:** 2
**Rating:** 4
**Confidence:** 3

**Summary:**

The authors generalise RoPE to a mechanism allowing it to choose angles in a way that is input dependent. The authors perform analysis mainly on linear-attention models and show that selective rope seems to improve performance over baselines such as RoPE or NoPE.

**Strengths:**

The connection between SSMs, linear attention, and RoPE is interesting. I particularly liked the presentation in Table 1.

**Weaknesses:**

My main area of research is in Transformers and not linear attention although I have some experience with linear attention and RFF.s

I do not think I quite understand the point of "Softmax attention implicitly applies a selective rotation, to encode relative positional information between tokens." Are you arguing that the rotations come from the relationship between the softmax kernel and RFF? So you can view the softmax kernel as applying RoPE but where the angles are sampled IID from a Gaussian. This however would really only be true if your angle samples tend to infinity of course. Is this how you are connecting RoPE with a "NoPE" softmax?

I found the notation slightly hard to follow especially as someone not coming from SMMs. I mainly found confusing that RoPE is really a method used in Transformers, but the paper seems to only implemented the selective mechanism for linear attention and was not implemented for normal quadratic Transformers? Is there something stopping you from implementing for a normal Transformer?

Minor
Typo in abstract: rotation in all angels -> rotation in all angles

**Questions:**

Please see questions in the weaknesses

---

> ### Author Response · Authors · 2025-11-21
>
> We thank the reviewer for their positive feedback and for finding our theorem and presentation interesting. Below, we clarify the question regarding the connection between softmax transformers and random rotations.
>
> **Question on softmax attention implicitly applies a selective rotation**
>
> The reviewer’s understanding is correct. Our theorem relies on approximating softmax using Random Fourier Feature (RFF) kernels and shows that softmax attention can be interpreted as applying random rotations, where the angles are sampled i.i.d. from a Gaussian distribution. In the limit as the number of random features goes to infinity, this approximation converges exactly to the softmax kernel. This interpretation motivates our design of **Selective RoPE** for linear transformers and SSMs, with the goal of boosting their performance and reducing the performance gap with standard softmax transformers. We also provide empirical evidence for this formulation to help improve the performance of softmax attention, which you can observe in the new version of the paper.
>
> **Applying Selective RoPE to softmax Transformer**
>
> We thank the reviewer for the thoughtful suggestion. We have applied Selective RoPE to the softmax transformer (w. Decay) and the results are shown below:
>
> |Model|LMB.(ppl↓)|LMB.(acc↑)|PIQA(acc↑)|Hella.(acc_n↑)|Wino.(acc↑)|ARC-e(acc↑)|ARC-c(acc_n↑)|Avg.|
> |-|-|-|-|-|-|-|-|-|
> |**Transformer** (w. Decay)||||||||
> |NoPE|26.04|37.4|*69.6*|47.0|**55.2**|50.7|*25.8*|47.6|
> |RoPE|*23.16*|*37.7*|69.5|*47.6*|*55.0*|**52.7**|25.3|*48.0*|
> |SelectiveRoPE| **21.89** | **38.2** | **70.2** | **47.8** | 54.1 | *52.4* | **26.1** | **48.1**|
>
> These results show that **Selective RoPE outperforms both standard RoPE and having no positional embedding (NoPE)**.
>
> Moreover, although RoPE is primarily used in softmax transformers, its origin lies in linear attention with an imaginary gating mechanism, as discussed in Section 2 of our paper and in the original RoPE paper [1]. Our goal is to show that Selective RoPE is rooted in softmax attention via the RFF connection, while also naturally extending to linear transformers through an input-dependent imaginary forget gate.
>
> **Minor Typo in abstract**
> Thank you for pointing out the type, we have fixed it. Generally, we have improved the clarity and readability of the manuscript as we have detailed in the general response to all reviewers. The reviewer will also find that we have unified the notation in the main body of the paper and rely on Transformer notation with $q,k,v$ instead of SSM notation.
>
> -----
>
> ### References
>
>
> [1] RoFormer: Enhanced Transformer with Rotary Position Embedding. Jianlin Su, Yu Lu, Shengfeng Pan, Ahmed Murtadha, Bo Wen, Yunfeng Liu. 2024 Neurocomputing

---

### Official Review · Reviewer_uVEP · 2025-10-31

**Soundness:** 3
**Presentation:** 3
**Contribution:** 3
**Rating:** 6
**Confidence:** 3

**Summary:**

This paper presents a version of RoPE with learned, input-dependent arbitrary rotations. Theoretical analysis provided indicates that softmax attention implicitly performs selective rotations, motiving the proposed architecture. Selective RoPE uses a learned linear projection and cumulative sum to produce input-dependent rotations. An analysis of diagonal SSMs is provided which shows distinct roles for the real and imaginary parts of the state matrix, motiving the incorporation of Selective RoPE with GLA to provide better memory. Experiments on language modeling and show improvements with Selective RoPE compared to RoPE and softmax attention in GLA models.

**Strengths:**

The paper gives a detailed theoretical justification for the architecture design. The analyses of softmax attention as implicit rotation and spectral leakage in SSMs may be useful to future work. Experimental evidence is provided to support claims.

**Weaknesses:**

The conclusion that softmax attention applies implicit selective rotation is based on the RFF approximation and additional normalization assumptions. The paper does not prove that the resultant normalized approximation converges in the limit to softmax attention, and so this analysis may be overstating the connection.


The real-data language modeling results in table 3 omit the RoPE condition.

**Questions:**

Did you compute RoPE setting for Table 3?

---

> ### Author Response · Authors · 2025-11-21
>
> We thank the reviewer for their thoughtful feedback and are glad that they found our theoretical insights valuable. Below, we address the main concern raised by the reviewer, along with their specific questions:
>
>
> **Concern: Clarifying the Assumptions Behind Implicit Selective Rotation in Softmax Attention**
>
> Thank you for your attention to detail. Referring to the derivation of the linear attention with rotation at Appendix A.2, we note that the equations (16), (17), and (18) result in a recurrence with the scalar term $\exp\left(\frac{\Vert \mathbf{q}\_t\Vert\_2^2 - \Vert \mathbf{q}\_{t-1}\Vert\_2^2}{2}\right)$ in the transition matrix along with the rotation matrix. Consequently, our assumption for a fixed norm for the query would allow for this term to vanish, resulting in the rotation matrix being the only component in the transition matrix. We have multiple reasons for why this assumption makes sense:
>
> 1. The normalization of queries and keys is actually a common practice in softmax attention. Namely, OLMO and Kimi-K2 both incorporate the QK-Norm idea introduced by [1] in their model for better stability. This idea was first introduced in [2].
>
> 2. There is evidence in the literature that the norm of the query has an extremely sharp distribution over sequence, especially when compared to the norm of the key [3]. In order to further test this claim, we have performed our own experiments, the results of which you can observe [here](https://drive.google.com/drive/folders/1Sm0KnlHpds_bUrKu3MSasTLpY3hN12yP?usp=sharing). In this experiment, we trained a 24 layer transformer with 24 heads (340M parameters) on 15B tokens, and plotted the norm of the query over a sequence of size 1024. As we observe, the norm of the query remains consistently in a small neighborhood, effectively looking constant.
> Regarding the norm of the key, from (18) we observe that we can absorb the norm of the key into the value vector $\mathbf{v}_t$. Consequently, the assumption over the norm of the key was solely made for the purpose of keeping the mathematical derivation simple and easy to follow.
>
>
> **Question: Adding RoPE to Table 3**
> We thank the reviewer for their suggestion. We have included RoPE as a baseline in our language modeling experiments and show that **Selective RoPE outperforms** standard RoPE for GLA.
>
> |Model|LMB.(ppl↓)|LMB.(acc↑)|PIQA(acc↑)|Hella.(acc_n↑)|Wino.(acc↑)|ARC-e(acc↑)|ARC-c(acc_n↑)|Avg.|
> |-|-|-|-|-|-|-|-|-|
> |**GLA**|||||||||
> |NoPE|**19.21**|**39.4**|*69.7*|**48.0**|53.1|50.9|24.6|*47.6*|
> |RoPE|23.96|36.1|*69.7*|47.7|**54.0**|50.9|*25.1*|47.2|
> |Selective RoPE|*21.16*|*37.4*|**70.6**|*47.9*|*53.9*|**52.0**|**26.2**|**48.0**|
>
> Moreover, we have extended our experiments to include Gated DeltaNet and Transformer (with decay), and demonstrate that Selective RoPE consistently outperforms RoPE across all these models as shown in **general response to all reviewers**, highlighting its broader applicability and potential.
>
>
> --------
> ### References
> [1] Scaling Vision Transformers to 22 Billion Parameters. M. Dehghani, J. Djolonga, B. Mustafa, P. Padlewski, J. Heek, J. Gilmer, A. Steiner, M. Caron, R. Geirhos, I. Alabdulmohsin. (2023) ICML 2023.
>
> [2] Query-Key Normalization for Transformers. A. Henry, P. R. Dachapally, S. S. Pawar, and Y. Chen. (2020) Findings of EMNLP 2020.
>
> [3] TRAMS: Training-free Memory Selection for Long-range Language Modeling. H. Yu, C. Wang, Y. Zhang, and W. Bi. (2023) Findings of EMNLP 2023.

---

### Official Review · Reviewer_zi2k · 2025-11-01

**Soundness:** 2
**Presentation:** 2
**Contribution:** 2
**Rating:** 4
**Confidence:** 3

**Summary:**

The paper proposes a selective Rotary Position Embedding (RoPE) that uses an input-dependent rotation to enhance the performance of models with Gated Linear Attention. The paper provides a theoretical analysis of how softmax attention performs a hidden form of rotation and further proposes to determine the rotation angle via a linear projection of the query. Experiments are conducted on GLA showing that selective RoPE achieves better performance than NoPE and RoPE.

**Strengths:**

* The paper is clear with summarized insights and clear figures.

* The analyses on the implicit rotation of softmax attention are interesting.

* The paper provides an in-depth analysis from the RFF perspective.

**Weaknesses:**

* While the paper takes a lot of effort in the derivation of implicit selective rotation in softmax attention, the proposed method is applied to gated linear attention. Given that the derivation heavily relies on the Random Fourier Features (RFF) approximation, the practical impact of the proposed method has not been validated.

* Limited experiments. The experiments are conducted on small-scale models, which raises my concern about the stability and scalability of the proposed method. The leaned rotation angle may lead to unstable training.

**Questions:**

* It would be appreciated if the authors could provide additional experimental results of applying selective RoPE to softmax attention with more details on the experiments.

* Since rotations are composable. Can the proposed method be equivalently viewed as applying a rotation to the queries and keys before the RoPE operation? What is the significance of doing so?

---

> ### Author Response · Authors · 2025-11-21
>
> We thank the reviewer for their thoughtful feedback and are pleased that our insights, analysis of softmax attention, and in-depth theoretical contributions resonated with them. Below, we address the concerns raised in the review:
>
> **1) Training stability of Selective RoPE:** We agree with the reviewer regarding their concerns about the training stability of Selective RoPE. This is in line with our observations that a naive application of Selective RoPE can lead to instabilities at higher learning rates where they are otherwise not observed for models using NoPE or RoPE at the same learning rate. This can indeed be traced back to the difficulties of learning functions with high-frequency components using gradient descent and has been studied in the literature [3, 4]. To remedy these issues we introduced a gating term on the rotary component (*phase gate* in the manuscript) and a weight normalization on the input projection for Selective RoPE. We also found that QK-norm [5] applied *after* Selective RoPE helps training stability significantly. Adding these normalization components has remedied all of our observed stability issues. We are currently running larger scale experiments (1.3B models) and will update our response with the stability results at this scale once the training runs finish. We expect these results to become available within the next week.
>
> **2) Applying Selective RoPE to softmax Transformer:**  We thank the reviewer for their thoughtful suggestion to improve the breadth of our empirical results. We applied Selective RoPE to the Softmax Transformer (with Decay) [1]. Our results show that Selective RoPE significantly improves the performance of the Softmax Transformer (w/ Decay) compared to both RoPE and No Position Embedding (NoPE). Additionally, we incorporated GLA with RoPE and Gated DeltaNet [2] as new state-of-the-art baselines, and we find that Selective RoPE provides performance gains for these models as well. Results are provided in **Table as our general response to all reviewers.**
>
>
> **3) Experiment on Larger scale models:** We have added two new baselines—Softmax Transformer w/ Decay and Gated DeltaNet—as state-of-the-art representations of sequence models. For all models, we evaluate RoPE, Selective RoPE, and No Position Embedding (NoPE) as consistent baselines, across which Selective RoPE shows superior performance. As mentioned in the previous point, we are currently training models at the *1.3B scale* and expect to report results within the coming week.
>
> ---
> ### References
>
> [1] Forgetting Transformer: Softmax Attention with a Forget Gate. Lin, Z., Nikishin, E., He, X. O., & Courville, A. (2025). ICLR 2025.
>
>
> [2]  Gated Delta Networks: Improving Mamba2 with Delta Rule. Yang, S., Kautz, J., & Hatamizadeh, A. (2025) ICLR 2025.
>
> [3] On the spectral bias of neural networks. N. Rahaman, A. Baratin, D. Arpit, F. Draxler, M. Lin, F. Hamprecht, Y. Bengio, and A. Courville. (2019) ICML 2019
>
> [4] Towards a Mathematical Theory of Super-resolution. E. Candès, C. Fernandez-Granda. (2014) Communications on Pure and Applied Mathematics
>
> [5] Query-Key Normalization for Transformers. A. Henry, P. Dachapally, S. Pawar, Y. Chen. (2020) Findings of EMNLP 2020

---

### Official Review · Reviewer_NYgW · 2025-11-01

**Soundness:** 2
**Presentation:** 3
**Contribution:** 2
**Rating:** 4
**Confidence:** 2

**Summary:**

The paper introduces Selective Rotary Position Embedding (Selective RoPE), an input-dependent mechanism designed to generalize standard Rotary Position Embeddings (RoPE) by performing rotations at arbitrary, selective frequencies.

**Strengths:**

Interesting theoretical insights, such as:
- Softmax attention implicitly applies a selective rotation, to encode relative positional information between tokens
- Linear Transformers can be enhanced by using both forgetting via real decay and rotation via imaginary gate.

**Weaknesses:**

Very limited evaluation for language modeling.  Would be good to have at least RoPE as baseline (Table 3) and evaluate Selective RoPE in different settings (i.e context length)

**Questions:**

Do you expect GLA  and Softmax Transformers to benefit equally from Selective RoPE?

---

> ### Author Response · Authors · 2025-11-21
>
> We thank the reviewer for their feedback and for considering our theoritical results **insightful and interesting**. Below we reply to the reviewers concern about limited language modeling experiments:
>
>
> We conducted extensive evaluations of Selective RoPE on both the Softmax Transformer (w/ Decay) [1] and Gated DeltaNet [2], a state-of-the-art linear transformer. For all models, we also evaluated RoPE and NoPE as baselines. Across every architecture Selective RoPE consistently improves perplexity (PPL) and downstream language-modeling benchmark performance.
>
> Results for the 370M-parameter Transformer, Gated DeltaNet, and GLA models, each evaluated with RoPE, Selective RoPE, and no positional embedding (NoPE), are summarized in the **table included in the general response** to all reviewers, which clearly shows Selective RoPE's superiority against other methods in all models.
>
> We are in the process of evaluating Selective RoPE on 1.3B parameter versions of the considered models. We are also running evaluations at different context lengths and will include the results for the latter together with the 1.3B results. We ask the reviewers for some time since these results are very compute intensive and difficult to realize in an academic setting. We expect the results to be ready within the next week and will update our responses accordingly.
>
>
> ------
> ### References
>
> [1] Forgetting Transformer: Softmax Attention with a Forget Gate. Lin, Z., Nikishin, E., He, X. O., & Courville, A. (2025). ICLR 2025.
>
> [2]  Gated Delta Networks: Improving Mamba2 with Delta Rule. Yang, S., Kautz, J., & Hatamizadeh, A. (2025) ICLR 2025.

---

### Author Response · Authors · 2025-11-21
**General Response to the Reviewers**

## General Response

We thank all the reviewers for their constructive feedback on our manuscript. As requested by several reviewers, we have extensively expanded our language-modeling experiments by including Gated DeltaNet (GDN) and the Softmax Transformer (w/ Decay). We compare Selective RoPE against RoPE and No Position Embedding across all models—GLA, GDN, and the Transformer w/ Decay (FoX)—and show that Selective RoPE consistently and significantly boosts performance for all models.

|Model|LMB.(ppl↓)|LMB.(acc↑)|PIQA(acc↑)|Hella.(acc_n↑)|Wino.(acc↑)|ARC-e(acc↑)|ARC-c(acc_n↑)|Avg.|
|-|-|-|-|-|-|-|-|-|
|**GLA**|||||||||
|NoPE|*23.15*|**39.4**|*69.7*|**48.0**|53.1|*50.9*|24.6|*47.6*|
|RoPE|23.96|36.1|*69.7*|47.7|**54.0**|*50.9*|*25.1*|47.2|
|Selective RoPE|**21.16**|*37.4*|**70.6**|*47.9*|*53.9*|**52.0**|**26.2**|**48.0**|
|**Gated DeltaNet**|||||||||
|NoPE|22.50|37.2|**70.9**|*47.6*|53.2|*52.0*|**25.9**|47.8|
|RoPE|*20.84*|*38.9*|*70.7*|**48.2**|*53.4*|51.3|25.1|*48.0*|
|Selective RoPE|**19.28**|**39.4**|70.1|*47.6*|**54.9**|**52.4**|*25.4*|**48.3**|
|**Transformer** (w/ Decay)||||||||
|NoPE|26.04|37.4|*69.6*|47.0|**55.2**|50.7|*25.8*|47.6|
|RoPE|*23.16*|*37.7*|69.5|*47.6*|*55.0*|**52.7**|25.3|*48.0*|
|Selective RoPE|**21.89**|**38.2**|**70.2**|**47.8**|54.1|*52.4*|**26.1**|**48.1**|

Moreover, we have included several new experimental results:

- An **ablation of the additional architectural components introduced in Selective RoPE** (phase gate and bias) on the MAD benchmark, showing that the core Selective RoPE mechanism already achieves  gains over NoPE/RoPE and that the variant with both phase gate and bias attains the best overall MAD average while preserving improvements across all MAD tasks.
- A corresponding ablation of these components in the language-modeling setup, where we systematically vary the presence of the phase gate and bias for GLA, GDN, and the Transformer w/ Decay (FoX in the manuscript); this confirms that Selective RoPE is robust across architectures, that the phase gate mainly helps optimization stability and downstream accuracy, and that adding only a bias does not yield consistent additional gains.
- An **efficient Triton implementation of Selective RoPE**, demonstrating that our fused kernel is almost as fast as RoPE/NoPE in prefill throughput and achieves up to a 3.4× speedup over the PyTorch-compile implementation at long sequence lengths, thereby addressing concerns about the runtime overhead of Selective RoPE. We will publish our implementation after the closure of the review phase.


We would also like to note that we have considerably improved the writing and general presentation of the manuscript. The content of the paper has remained unchanged but the readability has improved significantly. The primary changes are:

1. Consolidating the motivation for Selective RoPE and the description of the method in Section 3 (before: Sections 3 through 5).
2. Moving implementation details into their own subsection in the prelude to the experiments in Section 4.
3. Addition of a related work section (Section 5)

---

### Meta-Review · Area_Chair_f3Df · 2025-12-30

**Summary:**

This paper proposes Selective Rotary Position Embedding (Selective RoPE), an input-dependent generalization of RoPE motivated by a theoretical analysis that interprets softmax attention as implicitly performing selective rotations under an RFF approximation. The method is primarily instantiated in linear attention and SSM-style models (e.g., GLA, Gated DeltaNet), and extended to a softmax Transformer with decay. The paper aims to bridge positional encoding mechanisms across softmax attention, linear attention, and SSMs through a unified rotational perspective.

All reviewers acknowledge that the theoretical analysis is interesting and non-trivial, particularly the connections drawn between softmax attention, RFFs, implicit rotations, and the real/imaginary components of state transitions. Several reviewers note that these insights could be valuable beyond the specific method proposed.

The rebuttal substantially strengthens the empirical evaluation. In response to concerns about limited experiments and missing baselines, the authors add comprehensive language-modeling results across three architectures (GLA, Gated DeltaNet, and Transformer with decay), consistently comparing NoPE, RoPE, and Selective RoPE. These results show consistent but modest improvements in perplexity and downstream accuracy, addressing a major concern raised by multiple reviewers. The extension of Selective RoPE to a softmax-based Transformer also resolves ambiguity about whether the method is restricted to linear attention models.

However, one central concern remains insufficiently resolved. Multiple reviewers explicitly raised training stability and scalability issues, noting that learned, input-dependent rotations could introduce optimization instabilities, especially at higher learning rates or larger scales. In the rebuttal, the authors acknowledge these issues and state that stability problems exist without additional mechanisms (phase gate, normalization, QK-norm), and that these additions “remedy” the observed instabilities. While the authors provide qualitative explanations and reference prior work on spectral bias, the paper does not present a clear ablation or quantitative analysis that isolates stability effects, such as learning-rate sensitivity curves, failure cases, or comparisons showing when and why instability arises and how each component addresses it. The promised large-scale (1.3B) results and stability analyses are not yet available, leaving this concern partially speculative.

Overall, the work presents a conceptually interesting and reasonably well-executed idea, and the rebuttal meaningfully improves the experimental coverage. At the same time, the lack of a clean, explicit ablation focused on optimization stability and the reliance on assumptions in the softmax–RFF connection prevent the paper from fully substantiating its claims at scale. I would love to give a borderline accept with conditioned that the authors should clearly elaborate the stability ablation in the paper.

**Reviewer Concerns:**

The rebuttal and revised manuscript addressed most of the major concerns raised by the reviewers. The primary issue, shared by several reviewers, was the limited empirical evaluation and the absence of RoPE baselines. This was directly resolved through substantially expanded experiments that include RoPE and No Position Embedding across Gated Linear Attention, Gated DeltaNet, and softmax transformers with decay. These results consistently show that Selective RoPE improves performance, addressing concerns about empirical validity and generality.

Questions about whether Selective RoPE applies beyond linear attention and state-space models were also addressed. The authors implemented and evaluated the method on softmax transformers with decay, demonstrating clear gains over both RoPE and NoPE. This resolves earlier ambiguity about the scope of the method.

Concerns regarding the evaluation of training stability of large models from learned, input-dependent rotations still remained, while the authors provided a reasonable theoretical motivation and empirical evidence that these additions stabilize training, though very large-scale results are still in progress.

Some concerns remain partially outstanding. The interpretation that softmax attention implicitly performs selective rotations still relies on Random Fourier Feature approximations and normalization assumptions rather than a formal equivalence, and large-scale validation is not yet complete. These issues are now clearly scoped and do not substantially weaken the overall contribution.

**Reviewer Scores:**

Reviewer NYgW (initial: 4):
Likely to increase to 6, given that the main concern about limited language modeling evaluation and missing RoPE baselines was comprehensively addressed.

Reviewer zi2k (initial: 4):
Likely stay at 4, as stability concerns and some reservations about theory and scale may remain.

Reviewer uVEP (initial: 6):
Likely to remain at 6, as their primary concern about missing RoPE baselines and clarification of assumptions was resolved, even if some theoretical caveats persist.

Reviewer Rgdu (initial: 4):
Likely to increase to 6, given improved clarity, explicit softmax experiments, and better positioning of the RFF-based interpretation.

---

### Decision · Program_Chairs · 2026-01-26

Accept (Poster)